# Regression-Based Estimation of Causal Effects in the Presence of Selection Bias and Confounding

## Abstract

We consider the problem of estimating the expected causal effect $E[Y|do(X)]$ for a target variable $Y$ when treatment $X$ is set by intervention, focusing on continuous random variables. In settings without selection bias or confounding, $E[Y|do(X)] = E[Y|X]$, which can be estimated using standard regression methods. However, regression fails when systematic missingness induced by selection bias, or confounding distorts the data. Boeken et al. [2023] show that when training data is subject to selection, proxy variables unaffected by this process can, under certain constraints, be used to correct for selection bias to recover $E[Y|X]$, and hence $E[Y|do(X)]$, reliably. When data is additionally affected by confounding, however, this equality is no longer valid. In this work, we consider a more general setting and propose a framework that incorporates both selection bias and confounding. Specifically, we derive theoretical conditions ensuring identifiability and recoverability of causal effects under access to external data and proxy variables. We further introduce a two-step regression estimator (TSR), capable of exploiting proxy variables to adjust for selection bias while accounting for confounding. We show that TSR coincides with prior work if confounding is absent, but achieves a lower variance. Extensive simulation studies validate TSR's correctness for scenarios that include both selection bias and confounding with proxy variables.

## 1 Introduction

Recovering causal effects under selection bias is a fundamental challenge in empirical research. Specifically, we aim to estimate $E[Y \mid do(X)]$, the causal effect of a continuous treatment $X$ on a continuous target variable $Y$, from observational data that may be affected by selection mechanisms and confounding. Selection bias arises when the observed data fails to accurately represent the population due to preferential exclusion or conditioning on colliders, while confounding distorts the true causal relationships through (unobserved) common causes. Both phenomena are pervasive in real-world datasets and, if left unadjusted, can give rise to misleading conclusions.

Selection bias is a critical challenge in many real-world domains, including medicine [Berkson, 1946], economics, and machine learning, with recent examples highlighting its role in COVID-19 research [Herbert et al., 2020, Zhao et al., 2021], cancer progression modeling [Schill et al., 2024], and fairness in machine learning [Wang and Singh, 2021, Goel et al., 2021]. As a running example, consider that in loan risk assessment, banks may wish to isolate the causal effect of income ($X$) on loan default ($Y$) from other risk factors. Naturally, such a dataset only includes cases where loans have been issued ($S = 1$), introducing selection bias, as illustrated in Figure 1. Furthermore, unobserved factors like financial literacy of an individual may act as confounders that simultaneously influence income and loan default rates. Without proper adjustment for these biases, estimates of risk factors may be unreliable or even contradictory.

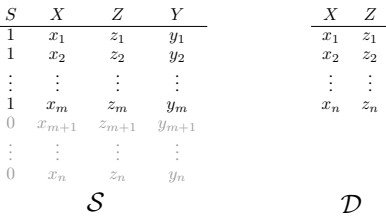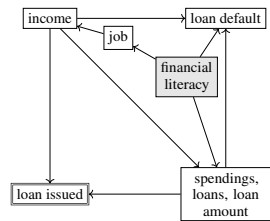

Figure 1: Left: Dataset composition: $\mathcal{S}$ is affected by a selection mechanism, $\mathcal{D}$ contains unbiased data. Right: Example applicable to our framework (covered by Assumption 2.3). When estimating the causal effect of income $X$ on loan default $Y$, the node "loan issued" represents the selection variable $S$, "financial literacy" is an unobserved confounder, and all other nodes serve as proxies ($Z$).

While identifiability of causal effects under confounding has been extensively studied, *recoverability* from selection-biased data has received comparatively less attention. Pearl's do-calculus provides a systematic framework for determining whether causal effects are identifiable under certain assumptions encoded in causal directed acyclic graphs (DAGs) [Pearl, 2009]. Building on this foundation, researchers explored recoverability from selection-biased data (s-recoverability) [Pearl, 2012, Correa et al., 2018, Jung et al., 2024, Mohan and Pearl, 2021]. Recent work by Boeken et al. [2023] emphasizes the importance of proposing practical estimators alongside identification results. They introduced regression-based methods to estimate $E[Y|X]$ for continuous targets assuming access to a proxy $Z$ for the selection variable, which renders the target $Y$ independent of the selection variable $S$ when conditioning on $\{X, Z\}$, i.e., $Y \per\!\!\!\perp S \mid \{X, Z\}$. Akin to other works on recoverability [Correa et al., 2018], they assume access to external data for $X, Z$ unaffected by selection (cf. Figure 1).

**Contributions**  We derive theoretical results ensuring identifiability and s-recoverability of causal effects with access to proxy variables and external data in Section 2.3. In the most general case, our results cover the graph shown in Figure 1, in which we can recover the causal effect of income on loan default by using the covariates spendings, etc. as a proxy for the selection variable. To account for the unobserved confounder "financial literacy", we leverage the information about the job type of an individual. We show that this setting is distinct from assumptions derived in prior work [Bareinboim et al., 2014, Correa et al., 2018], and propose a two-step regression estimator (TSR) based on our identification results in Section 3. Further, in Section 3.1, we analyze the bias and variance of TSR for the case in which confounding is absent, i.e., $E[Y|do(X)] = E[Y|X]$, and show that TSR is more efficient than the repeated regression estimator considered by Boeken et al. [2023]. We confirm those results, as well as the admissibility and usability of our estimator considering ordinary least squares (OLS) and ridge estimation in simulation studies in Section 4. We review closely related work in the corresponding sections and provide a more detailed discussion in Appendix A.1.

## 2 Recoverability and Identifiability

In the following, we consider causal effect estimation for a continuous target variable in the presence of selection bias and confounding, as illustrated in Figure 1. Before that, we outline the connection between missingness and selection bias, and introduce relevant notation and definitions to define recoverability from selection bias in Section 2.1. In Section 2.2, we review recoverability from selection bias without confounding, as studied by Boeken et al. [2023]. Subsequently, in Section 2.3, we derive a set of assumptions which ensure that the *causal effect is identifiable and recoverable from selection-biased data*. All proofs are provided in Appendix A.5.

### 2.1 Preliminaries and Notation

Throughout this paper, we follow the notational conventions introduced by Pearl [2009]. We consider a *causal directed acyclic graph* (DAG) model $(\mathcal{G}, P)$, where $P$ defines a distribution over the set of random variables $V$, which factorizes according to $\mathcal{G}$, and is consistent under interventions. $V$ includes all variables of interest ($\{Y, X, Z\} \subset V$) except for the binary *selection variable* $S$, where $S = 1$ denotes selection. That is, under selection bias, we only observe $P(V \mid S = 1)$. Further, $Y \in \mathbb{R}$ denotes the one-dimensional continuous *target* random variable, $X \in \mathbb{R}^p$, with $p \geq 1$, the potentially multidimensional and continuous random vector of *covariates*. Our interest lies in

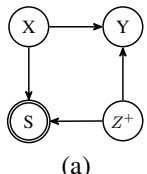 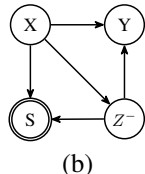 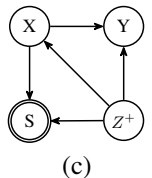 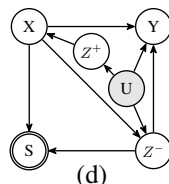

| (a) | (b) | (c) | (d) |

Figure 2: Example graphs that are consistent with Assumption 2.3. For graphs (a) and (b), we only need to adjust for selection bias since $E[Y \mid X]$ coincides with $E[Y \mid do(X)]$, while for graphs (c) and (d), adjustments for selection bias and confounding, as described in Section 2.3, are required.

estimating the *expected causal effect* $E[Y \mid do(X)]$, denoting the expected value of $Y$ when $X$ is set to a specific value by intervention (*hard or surgical intervention*). The potentially multidimensional and continuous random vector $Z \in \mathbb{R}^d$, with $d \geq 0$, may include confounding variables and *proxy variables* for the missing observations.

**Missingness and Selection Bias** *Selection bias* is typically induced by preferential selection and can be described as systematic missingness [Correa et al., 2019], also know as *missing not at random* (MNAR), where generally, $E[Y \mid X] \neq E[Y \mid X, S = 1]$. Therefore, proper adjustment is necessary when aiming to estimate $E[Y \mid X]$ or $E[Y \mid do(X)]$ from data affected by such systematic missingness. To approach this problem, we need to state some assumptions about the missingness scenario [Little and Rubin, 2002].

In the following, we distinguish between two missing data settings illustrated in Figure 1 (left), that are consistent with prior work [Boeken et al., 2023]. For both settings, we have independent and identically distributed (i.i.d.) observations of $(X, Y, Z) \sim P(X, Y, Z \mid S = 1)$ with index set $\mathcal{S}$ indicated by $S = 1$, where $P(X, Y, Z \mid S = 1)$ denotes the joint distribution of $X$, $Y$ and $Z$ conditioned on $S = 1$. Additionally, we observe realizations of i.i.d. $(X, Z) \sim P(X, Z)$ with index set $\mathcal{D}$ not underlying a selection process. In the first setting, the selected sample is a subset of the data not underlying the selection process ($\mathcal{S} \subset \mathcal{D}$). So, for $S = 0$, only the observations of $Y$ are missing. For the second setting, the selection bias setting, we have access to *external data* $\mathcal{D}$ for which $\mathcal{S} \cap \mathcal{D} = \emptyset$. Following the notation from Bareinboim et al. [2014], we call the unbiased data in both cases external data. If not stated otherwise, our results derived below hold for both settings.

**Recoverability** Before introducing a practical estimator, we need to ensure that the causal effect - which we are interested in - is recoverable [Pearl, 2009] from the available data. For cases involving selection bias, Pearl [2012] first proposed the concept of *s-recoverability*, which was later developed and defined as in Definition A.2 in Appendix A.3 by Bareinboim et al. [2014]. As there are settings that are only s-recoverable under consulting external unbiased data ($\mathcal{D}$) - which is the setting we focus on - Bareinboim et al. [2014] formulated a compatible definition for this case, restated below.

**Definition 2.1.** Given a causal DAG model $(\mathcal{G}, P)$ augmented with a node $S$, the distribution $Q = P(Y \mid X)$ is said to be *s-recoverable* from selection bias in $\mathcal{G}$ with external information over $T \subset V$ and selection-biased data over $M \subset V$ if the assumptions embedded in the causal model render Q expressible in terms of $P(M = m \mid S = 1)$ and $P(T = t)$, both positive.

Based on the notation and the definitions introduced above, we will now review the setting that was studied by Boeken et al. [2023], with selection bias induced by systematic missingness for which $P(Y \mid X)$ is s-recoverable when observing *privileged* information.

## 2.2 Recovering from Selection Bias in the Absence of Confounding

To recover from selection-biased data, Boeken et al. [2023] presented a special case of MNAR, the concept of *privilegedly missing at random* (PMAR), which is also known as the terminology of comparability in literature [Singh and Zhou, 2022]. In comparison to our work, this line of research assumes to have access to interventional/experimenta data. Miao et al. [2024] cover a different MNAR case than PMAR. However, their framework does not require external data. PMAR describes cases in which the target variable $Y$ is stochastically independent from the selection variable $S$, when conditioning on the covariates $X$ and proxy variables $Z$, as formalized in the assumption below.

**Assumption 2.2** (PMAR). Given a privilegedly observed set of variables $Z$, $Y$ is privilegedly missing at random (PMAR) if $S \perp\!\!\!\perp Y \mid \{X, Z\}$.

Intuitively, PMAR holds when treatment and proxies block all paths between target and selection variable, as in the DAG in Figure 2 (a). Under Assumption 2.2, by law of total expectation, $E[Y \mid X] = E[E[Y \mid X, Z] \mid X] = E[E[Y \mid X, Z, S = 1] \mid X]$. As discussed by Boeken et al. [2023], concerning discrete variables under certain positivity assumptions, s-recoverability is satisfied when selection biased $P(X, Y, Z \mid S = 1)$, as well as unbiased external $P(X, Z)$, are available and PMAR holds. They proposed a repeated regression (RR) for estimating $E[Y \mid X]$. The first regression models $E[Y \mid X, Z, S = 1]$ using $\mathcal{S}$. Next, predictions $\widetilde{Y} := \widehat{E}[Y \mid X, Z, S = 1]$ based on the first regression are computed on population-level in dataset $\mathcal{D}$, which is unaffected by selection. A second regression models $E[\widetilde{Y} \mid X]$ using $\mathcal{D}$, yielding the final estimate $\widehat{\mu}_{RR}(x) = \widehat{E}[\widetilde{Y} \mid X]$ for $E[Y \mid X]$.

Besides repeated regression, Boeken et al. [2023] also proposed an estimator based on inverse probability weighting and a doubly robust estimator. However, repeated regression was the clear favorite in their evaluation, which is why we omit the other two estimators.

In absence of confounding between $X$ and $Y$, this method also provides a reliable estimate for $E[Y \mid do(X)]$ as $E[Y \mid do(X)] = E[Y \mid X]$ can be ensured. This is the case in Figure 2 (a) and (b). However, as soon as we flip the edge between $X$ and $Z$ in Figure 2 (b), arriving at the graph in Figure 2 (c), we additionally need to adjust for confounding. In the following section, we discuss this issue in more detail, and propose a solution for such settings.

## 2.3 Identification under Selection Bias in Presence of Confounding

Criteria for causal effect identification and s-recoverability under confounding and selection bias have been generalized by multiple authors [Pearl, 2012, Bareinboim and Tian, 2015, Correa et al., 2018, 2019]. To treat both sources of bias, they propose to decompose $Z = Z^+ \cup Z^-$ into $Z^+$, the set of the non-descendants of $X$ and $Z^-$, the set of descendants of $X$ that are included in $Z$. Based on this distinction, Bareinboim et al. [2014] introduced the *selection backdoor criterion* (provided in Assumption A.3) under which the causal effect is identifiable and s-recoverable. We adjust the assumptions proposed by Bareinboim et al. [2014], as stated in Assumption 2.3 below, to ensure identifiability and s-recoverability of the causal effect for PMAR with potentially unobserved confounding. An example graph, which is not covered by previous approaches is shown in Figure 2 (d), where $U$ is not included in $Z$ and may be an unobserved confounder. We compare our assumptions with prior works [Bareinboim et al., 2014, Correa et al., 2018] in Appendix A.4.

**Assumption 2.3.** Decompose the set of variables $Z$ into $Z = Z^+ \cup Z^-$, where $Z^+$ are non-descendants of $X$ and $Z^-$ are descendants of $X$. Assume that

1. $X$ and $Z$ block all paths between $S$ and $Y$, namely $S \perp\!\!\!\perp_{\mathcal{G}} Y \mid \{X, Z\}$ (PMAR)

2. $Z^+$ blocks all backdoor paths between $X$ and $Y$, namely $Y \perp\!\!\!\perp_{\mathcal{G}_{\underline{X}}} X \mid Z^+$

3. $Z \cup \{X, Y\} \subset M$, where variables $M$ are collected under selection bias (dataset $\mathcal{S}$) and $Z \subset T$, where $T$ is collected on population-level (dataset $\mathcal{D}$). If $Z^- \neq \emptyset$, $X \subset T$.

We illustrate the assumptions shortly. Note that whenever Assumption 2.3 is satisfied and $Z^+ = \emptyset$, confounding is excluded and we recover the setting from Boeken et al. [2023]. When $Z^+$ and $Z^-$ are present, $Z^+$ shields the confounding of $X$ and $Y$, and both $Z^-$ and $Z^+$ are needed to adjust for the selection bias. In case of $Z^- \neq \emptyset$, we have to also observe $X$ unbiased.

For instance, in Figure 1, we may have additional measurements from an unbiased source of $X = \{income\}$ and $Z = \{job, spending, loans, loan amount\}$, for which the label is not available, e.g., because collecting the label is costly. In Appendix A.4, we will elaborate more on limited access to unbiased data. When Assumption 2.3 is satisfied, the causal effect is identifiable and s-recoverable:

**Theorem 2.4.** *Under Assumption 2.3, the causal effect $E[Y \mid do(X)]$ is identifiable, s-recoverable and can be expressed as follows*

$$\int_{z^+} E[E[Y \mid X, Z^+ = z^+, Z^-, S = 1] \mid X, Z^+ = z^+] P(Z^+ = z^+) dz^+ \,.$$

Based on the identification result above, we develop a practical estimator for continuous targets. For notational reasons, we propose an estimator for linear cases and explain its extension to the non-linear setting in Section 3.2. Assumption 2.5 contains the linearity assumptions required for our estimator.

**Assumption 2.5.** Let any observation of $Y$ be defined through the following assignment:

$$y := \beta_0 + \beta_1 x + \beta_2 z^+ + \beta_3 z^- + \epsilon \,,$$

where $(x, y, z)$ are drawn i.i.d. from $P(X, Y, Z)$ and $\epsilon$ is drawn i.i.d. from a standard normal distribution $\mathcal{N}(0, 1)$. The coefficients $\beta_0$, $\beta_1$, $\beta_2$ and $\beta_3$ are of the dimension of its corresponding vector of variables $X \in \mathbb{R}^p$, $Z^+ \in \mathbb{R}^{d_1}$ or $Z^- \in \mathbb{R}^{d_2}$ respectively.

In summary, we assume a linear setting with Gaussian error terms, as common in regression. Based on Assumption 2.5, Theorem 2.4 simplifies to Theorem 2.6. Corollary 2.7 further simplifies the expression under certain conditions such that no integral calculation needs to be carried out.

**Theorem 2.6.** *Under Assumption 2.3 and Assumption 2.5, the causal effect $E[Y \mid do(X)]$ is identifiable, s-recoverable and can be expressed as*

$$E[Y \mid do(X = x)] = \beta_0 + \beta_1 x + \beta_2 E[Z^+] + \beta_3 \int_{z^+} E[Z^- \mid X = x, Z^+ = z^+] P(Z^+ = z^+) dz^+. \tag{1}$$

**Corollary 2.7.** *If $Z^+$ blocks all backdoor paths between $X$ and $Z^-$, the integral from above reduces to $E[Z^- \mid do(X)]$. If additionally, $X$ and $Z^-$ are not confounded, it reduces to $E[Z^- \mid X]$.*

Based on the above results, we will introduce our a practical estimator in the next section.

# 3   A Two-Step Regression Estimator

Next, we derive an estimator for $E[Y \mid do(X)]$ by substituting each component of the causal effect expression in Equation (1) with its corresponding estimator. We refer to this estimator as the **Two-Step Regression Estimator (TSR)**. For a specific value of $x$, it is defined as

$$\hat{\mu}_{TSR}(x) = \boxed{\hat{\beta}_0} + \boxed{\hat{\beta}_1} x + \boxed{\hat{\beta}_2} \boxed{\widehat{E}[Z^+]} + \boxed{\hat{\beta}_3} \int_{z^+} \boxed{\widehat{E}[Z^- \mid X = x, Z^+ = z^+]} \boxed{\hat{P}(Z^+ = z^+)} dz^+ \,, \tag{2}$$

emp. mean
OLS: $Y \sim X, Z^+, Z^-$
OLS: $Z^- \sim X, Z^+$
density est.

where some components can be estimated based on the selected dataset $\mathcal{S}$, whereas parts of the estimates require access to an external dataset $\mathcal{D}$, which is not underlying the selection mechanism. In particular, we obtain the estimates $\hat{\beta}_0$, $\hat{\beta}_1$, $\hat{\beta}_2$, $\hat{\beta}_3$ by OLS for the model $E[Y \mid X, Z^+, Z^-, S = 1] = \beta_0 + \beta_1 X + \beta_2 Z^+ + \beta_3 Z^-$ based on the observations in $\mathcal{S}$ in the first step. In the second step, we estimate $E[Z^+]$ by its empirical mean and approximate the integral $\int_{z^+} E[Z^- \mid X = x, Z^+ = z^+] P(Z^+ = z^+) dz^+$ by OLS estimations of $E[Z^- \mid X = x, Z^+ = z^+]$ weighted by an estimation of the density $P(Z^+ = z^+)$ of $Z^+$, both based on observations from $\mathcal{D}$.

In the following, we discuss several possible instantiations of our estimator depending on whether or not the data is affected by confounding and whether or not certain sets of variables are empty. For all settings, we assume that the considered variables meet the assumptions required for Theorem 2.6.

In the *absence of confounding*, where $E[Y \mid X] = E[Y \mid do(X)]$ with $Z^+ = \emptyset$ and implicitly $Z = Z^-$, as in Figure 2 (b), TSR - coinciding with RR - reduces to (cf. Appendix A.6)

$$\hat{\mu}_{TSR}(x) = \hat{\beta}_0 + \hat{\beta}_1 x + \hat{\beta}_2 \widehat{E}[Z^- \mid X = x] \,.$$

In scenarios for which we cannot exclude *confounding* and thus $E[Y \mid do(X)] = E[Y \mid X]$ cannot be ensured, we distinguish two cases that are illustrated in Figure 2. First, consider the minimal example, presented in Figure 2 (c), with $Z = Z^+$ and $Z^- = \emptyset$. Here, the TSR estimator reduces to

$$\hat{\mu}_{TSR}(x) = \hat{\beta}_0 + \hat{\beta}_1 x + \hat{\beta}_2 \widehat{E}[Z^+] \,, \tag{3}$$

which differs from the RR estimation (see Appendix A.6). The second estimation step only requires calculating the empirical mean of $Z^+$. This is intuitively comprehensible because $E[Z^+ \mid do(X)] = E[Z^+]$ when $Z^+$ is a non-descendant of $X$. Hence, in this setting, only $Z^+$, but not $X$, needs to be observed in an external unbiased dataset.

For our running example in Figure 1 and Figure 2 (d), we have to compute the full estimator to recover the causal effect of $X$ on $Y$. Hence, the estimator is given by Equation 2. Whenever we can assume linearity $E[Z^- \mid X = x, Z^+ = z^+] = \gamma_0 + \gamma_1 x + \gamma_2 z^+$ with $\gamma_0 \in \mathbb{R}^{d_2}, \gamma_1 \in \mathbb{R}^{d_2 \times p}$ and $\gamma_2 \in \mathbb{R}^{d_2 \times d_1}$, TSR takes the form

$$\hat{\mu}_{TSR}(x) = \hat{\beta}_0 + \hat{\beta}_1 x + \hat{\beta}_2 \widehat{E}[Z^+] + \hat{\beta}_3(\hat{\gamma}_0 + \hat{\gamma}_1 x + \hat{\gamma}_2 \widehat{E}[Z^+]) \, , \tag{4}$$

where we impute the regression coefficient estimates of the regression of $Y$ on $X, Z^+, Z^-$ in $\mathcal{S}$ and of the regression of $Z^-$ on $Z^+$ and $X$ in $\mathcal{D}$, as well as the mean estimate of $Z^+$ in $\mathcal{D}$.

In addition to observed confounding, TSR can handle *unobserved confounding* between $X$ and $Y$ for cases in which $Z^+$ blocks all back-door paths between $X$ and $Y$ which arise through the confounder.

**Regularization** Note that since the variables in $X$ and $Z$ might be highly correlated in $\mathcal{S}$, it can be profitable to implement the first regression in TSR and RR with a ridge regression penalty, to reduce the variance of the estimator. In addition, even for the second estimation step, ridge regression in $\mathcal{D}$ should be considered because, for instance, in Figure 2 (d), $X$ and $Z^+$ are correlated. In our empirical evaluation in Section 4.1, we therefore also instantiate both TSR and RR with a ridge penalty.

## 3.1 Analysis of Bias and Variance

Next, we examine unbiasedness, derive the variance of the proposed two-step regression (TSR) estimator and compare it to the variance of the repeated regression (RR) estimator from Boeken et al. [2023] for graphs aligned with Figure 2 (a), where $E[Y \mid X] = E[Y \mid do(X)]$, $Z = Z^+$ and $Z^- = \emptyset$. For simplicity, we assume that all variables are univariate and linearly related, to get an intuition of the bias and variance of both estimators. We formalize our assumptions below.

**Assumption 3.1.** Let any observation $y$ be generated through the following assignment:

$$y := \beta_0 + \beta_1 x + \beta_2 z^+ + \epsilon \, ,$$

where $(x, z^+)$ are drawn i.i.d. from $P(X, Z^+)$. In particular, $z^+ = \mu_{z^+} + \xi$, with $\mu_{z^+} \in \mathbb{R}$, $\xi$ and $\epsilon$ are drawn i.i.d. from $\mathcal{N}(0, 1)$. Value $x$ and the coefficients $\beta_0, \beta_1, \beta_2$ are in $\mathbb{R}$.

In this simplified setting, TSR reduces to $E[Y \mid do(X = x)] = \beta_0 + \beta_1 x + \beta_2 \mu_{Z^+}$, where $\mu_{Z^+} = E[Z^+ \mid do(X = x)]$. Thus, the second step of the TSR estimator reduces to estimating the mean of $Z^+$. In contrast, RR performs an OLS estimate of $E[Z^+|X = x]$ in $\mathcal{D}$, which is given by $\widehat{E}[Z^+ \mid X = x] = \alpha_0 + \alpha_1 x$ with correct coefficients $\alpha_0 = E[Z^+]$ and $\alpha_1 = 0$. Assuming that in both regression steps, unbiased estimation is ensured, i.e., the chosen model class includes the ground truth generating mechanism, TSR and RR are ensured to be unbiased under $\mathcal{S} \cap \mathcal{D} = \emptyset$. In that case, the data points for both regression steps are independent of each other and therefore the coefficient estimators of the two steps are. In contrast, for $\mathcal{S} \subset \mathcal{D}$, the bias in point $x$ is $Cov[\hat{\beta}_2, \overline{Z^+}]$ for TSR and $Cov[\hat{\beta}_2, \hat{\alpha}_0 + \hat{\alpha}_1 x]$ for RR (see derivation in Appendix A.5). We also investigate this aspect empirically through simulation studies in Section 4.1, which suggest that the bias terms for $\mathcal{S} \subset \mathcal{D}$ might be negligible. Intuitively, the smaller the overlap of $\mathcal{S}$ and $\mathcal{D}$, the smaller the dependence between the estimates of those two samples. We discuss the bias in a more general case in Appendix A.5.

After studying the bias of both estimators, we now compare their variance. Here, we restrict ourselves to the case in which $\mathcal{S} \cap \mathcal{D} = \emptyset$, exploiting the independence of the observations of $\mathcal{S}$ and $\mathcal{D}$.

**Theorem 3.2.** *Under Assumption 3.1 and $\mathcal{S} \cap \mathcal{D} = \emptyset$, let $\widehat{E}[Z^+ \mid X = x] = \hat{\alpha}_0 + \hat{\alpha}_1 x$ be the OLS estimator of the second step for $\hat{\mu}_{RR}(x)$ and $\bar{X} = \frac{1}{|\mathcal{D}|} \sum_{i=1}^{|\mathcal{D}|} X_i$, then*

$$Var[\hat{\mu}_{RR}(x)] - Var[\hat{\mu}_{TSR}(x)] = Var[\hat{\beta}_2(\hat{\alpha}_1(x - \bar{X}))] \geq 0 \, .$$

The result implies that in the second regression step, the irrelevant regressor $X$ inflates the variance in small samples. The magnitude of the difference between the variances depends on the estimated

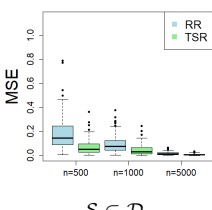 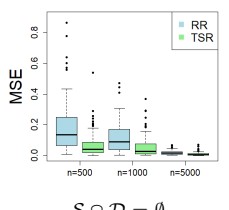 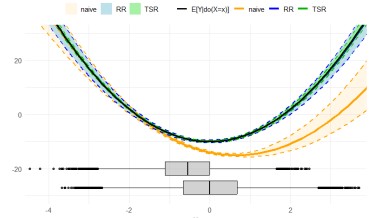

Figure 3: Left: Boxplots of the MSE over $\mathcal{D}$ for $\mathcal{S} \subset \mathcal{D}$, and $\mathcal{S} \cap \mathcal{D} = \emptyset$ (middle) of RR and TSR for $n \in \{500, 1000, 5000\}$. On the right, we show the associated 95%-areas of naive, RR and TSR for $n = 500$ (for $\mathcal{S} \cap \mathcal{D} = \emptyset$). The upper boxplot in the figure represents the distribution of $X$ in $\mathcal{S}$ and the lower in $\mathcal{D}$. The curves for RR and TSR display the mean estimation over all simulations.

effect of $Z$ on $Y$, the error in estimating $\alpha_1 = 0$ and the distance between $x$ and the empirical mean of the distribution of $X$ in $\mathcal{D}$. As $\hat{\alpha}_1$ converges to zero when the sample size of $\mathcal{D}$ goes towards infinity, the difference between the variances of RR and TSR at point $x$ also converges to zero.

The above results imply that we expect a lower mean squared error (MSE) for TSR than for RR, which we confirm by simulation-based experiments in Section 4.1. Subsequently, we empirically evaluate both estimators with a ridge regression penalty and observe that regularization can reduce the mean squared error by introducing some bias.

## 3.2  Introducing Non-linearity

In the previous sections, we outlined our theory for a linear estimator. As standard, we can extend the TSR estimator to non-linear settings by considering feature maps of the inputs. In particular, we can exchange $X$, $Z^+$ or $Z^-$ by vectors $\varphi_X(X)$, $\varphi_{Z^+}(Z^+)$, $\varphi_{Z^-}(Z^-)$ respectively, where $\varphi_X$, $\varphi_{Z^+}$, $\varphi_{Z^-}$ denote feature maps from a vector of variables to a vector of functions of the variables in $X$, $Z^+$ and $Z^-$ respectively. For example, in our experiments we perform polynomial regression. The linear case, can hence be seen as a special case with polynomials up to degree 1.

# 4  Experiments

In this section, we empirically evaluate the proposed Two-Step Regression (TSR) estimator, and compare it to Repeated Regression (RR) [Boeken et al., 2023]. We also instantiate both estimators with a ridge penalty in the regression based on $\mathcal{S}$ with penalization parameter $\lambda \in \{10^{-2}, 10^{-1.9}, ..., 10^2\}$, chosen via cross-validation. As a *naive* baseline, we consider the OLS regression estimator trained only on data from $\mathcal{S}$, which estimates $E[Y|X, S = 1]$ instead of $E[Y|X]$. We generate train and test data, both consisting of a selected dataset $\mathcal{S}$ and a population-level dataset $\mathcal{D}$. We chose the same sample size $n$ for $\mathcal{D}$ in both the test and the training data. The sample size for $\mathcal{S}$ is generated randomized by the selection process. All results are based on 100 simulation runs respectively.

First, in Section 4.1, we confirm our results for the variance comparison of TSR and RR from Section 3.1. Then, in Section 4.2, we look at several examples with confounding, for which RR is not applicable. Finally, we will evaluate the performance in a more challenging setting based on Figure 2 (d), in Section 4.3. Additional results are provided in Appendix B.

## 4.1  Empirical Variance Evaluation

As discussed in Section 3.1, the variance of TSR is at most of the same magnitude as of RR. The result was proven only for $\mathcal{S} \cap \mathcal{D} = \emptyset$. Here, we simulate data according to both settings, $\mathcal{S} \subset \mathcal{D}$ and $\mathcal{S} \cap \mathcal{D} = \emptyset$ and consider the causal effect $E[Y|do(X = x)]$ to be both a linear and a quadratic function in $x$. We generate data according to the DAG shown in Figure 2 (a), as follows:

$$X, \varepsilon_Y \sim \mathcal{N}(0, 1) \qquad\qquad Z^+ \sim \mathcal{N}(-2, 1) \qquad\qquad S := \mathbf{1}(X + Z^+ < -2) \, .$$

We use $Y_{\text{lin}} := 3X + 5Z^+ + \varepsilon_Y$ in the linear case, and replace $X$ with $X^2$ for the quadratic case. For the linear model, computing TSR and RR, we include regressors up to degree 1. In the quadratic model, we used regressors up to degree 2. We chose the sample size $n$ to be 500, 1000 and 5000.

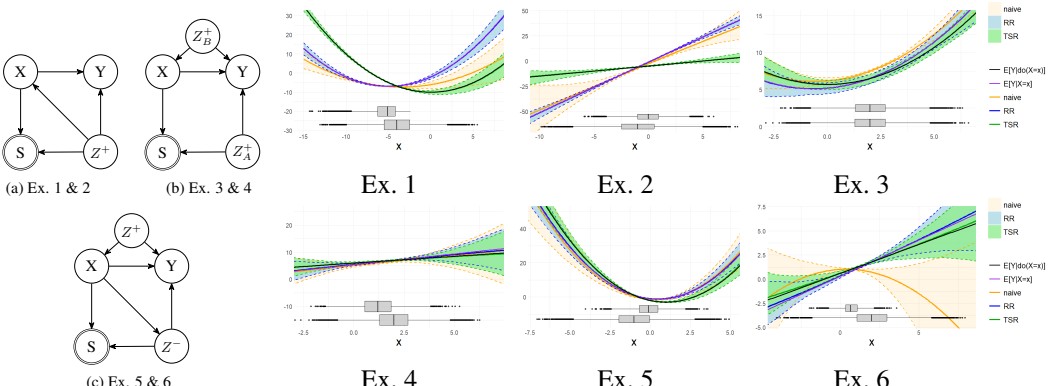

Figure 4: Left: DAGs representing the DGPs of Examples 1-6, Right: Comparison of the central 95%-areas of RR and TSR with OLS regression for $n = 1000$ in the setting with $\mathcal{S} \cap \mathcal{D} = \emptyset$ for Examples 1-6, respectively. The upper boxplot represents the distribution of $X$ in $\mathcal{S}$ and the lower in $\mathcal{D}$. The curves for RR and TSR display the mean estimation over the simulation runs.

The results for the quadratic model in Figure 3 show that the mean MSE of TSR is consistently lower than for RR, whereas the difference vanishes when increasing the sample size $n$ (details are provided in Table 1 in Appendix B.1), validating our theoretical results. The plot on the right shows the area in which the central 95% of the estimates of $E[Y|do(X)]$ from the 100 simulation runs (for $\mathcal{S} \cap \mathcal{D} = \emptyset$). We see that the estimates for TSR are more concentrated around the mean than for RR. As expected, the naive estimator is systematically biased.

We provide all numerical results, and compare the errors on test data sampled from $\mathcal{S}$ to data from $\mathcal{D}$, for linear and quadratic functions in both scenarios ($\mathcal{S} \cap \mathcal{D} = \emptyset$, and $\mathcal{S} \subset \mathcal{D}$) in Appendix B.1. Overall, the results shown in Appendix B.1 confirm the conclusions drawn from this section.

### 4.2 Simulations with Selection Bias and Confounding

Next, we consider in total six distinct generating mechanisms, as illustrated in Figure 4, that include confounding variables. Hence, RR is not applicable for estimating the causal effect since $E[Y \mid do(X)] \neq E[Y \mid X]$, but should be able to recover $E[Y \mid X]$. The details for the data generating processes are provided in Appendix B.2, where for each graph, we provide a linear and quadratic generative mechanism. Corollary 2.7 ensures that no integral has to be computed for TSR in the setting from Figure 4 (c). For Ex. 1 & 2, TSR is given by Equation (3). For Ex. 3 & 4, an additional summand for a second element in $Z^+$ is included, and for Ex. 5 & 6, we use Equation (4).

We included regressors up to degree 2, to match the generating mechanisms. In Figure 4, we plot the empirical central 95% confidence intervals for TSR, RR, with OLS regression and the naive baseline for the case of $n = 1000$ for each setting. Interestingly, RR and naive have the highest error for Examples 1 and 2 for which the underlying graph is depicted in Figure 4 (a), whereas the difference between TSR and RR is less pronounced for the other examples. In Example 6, the baseline exhibits a strong bias emphasizing the need for adjustment. We further report the results on $\mathcal{D}$ and $\mathcal{S}$ for OLS, and ridge regression for both TSR, and RR increasing sample size in Appendix B.2. Additionally, we report the numerical results of the errors on $\mathcal{S}$ and $\mathcal{D}$. In most cases, ridge regression matches the performance of OLS. In Example 1, however, we observe that ridge regression introduces a bias for both TSR and RR, suggesting that, in regions with low support (at the borders), the 95%-areas do not, or only barely include $E[Y|do(X)]$ for TSR and $E[Y|X]$ for RR, respectively.

### 4.3 Simulations with Selection Bias and Unobserved Confounding

Last, we consider a case with *unobserved* confounding. The data generating process is based on Figure 2 (d) which is the graph compatible with the motivating example in Figure 1:

$$U, \varepsilon_{Z^+}, \varepsilon_{Z^-}, \varepsilon_X, \varepsilon_Y \sim \mathcal{N}(0,1) \quad Z^+ := 2U + \varepsilon_{Z^+} \quad\quad S := \mathbf{1}(X + Z^- > 5)$$
$$X := Z^+ + \varepsilon_X \quad\quad\quad\quad\quad Z^- := X + 2U + 2\varepsilon_{Z^-} \quad Y := 0.5X^2 + 2Z^- + 2U + 3\varepsilon_Y.$$

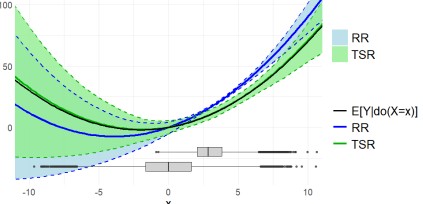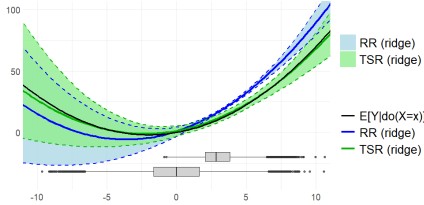

Figure 5: Comparison of the central 95%-areas of TSR and RR for the DAG in Figure 2 (d) with sample size $n = 500$ and $\mathcal{S} \cap \mathcal{D} = \emptyset$. The upper boxplot represents the distribution of $X$ in $\mathcal{S}$ and the lower in $\mathcal{D}$. The curves for RR and TSR display the mean estimation over the simulation runs.

As outlined in Section 3, in this case, TSR is given by Equation (4), where we restrict ourselves notational to the linear case. The $\beta$ coefficients are estimated by OLS from $\mathcal{S}$, whereas $\hat{\gamma}$ coefficients are estimated from $\mathcal{D}$. Further, $\widehat{E}[Z^+]$ is the empirical mean of $Z^+$ in $\mathcal{D}$. Akin to the previous experiments, we use OLS-based regression and optionally add a ridge penalty for the regressions. We explicitly add the penalty also to the second step since $X$ and $Z^+$ are correlated.

We show the results for $n = 500$ in Figure 5, where we observe that TSR is able to recover the ground truth. We additionally show RR as a baseline, but note that this setting violates its underlying assumptions. Hence, it is expected that it does not recover the ground truth causal effect. In addition, we observe that the confidence intervals for ridge are slightly smaller than for OLS, while a small bias is introduced. We repeat the experiment for $n = 2000$, for which we show the results in Appendix B.3, where we observe that the difference between OLS and ridge is not evident anymore.

## 5 Conclusion

We considered estimation of the causal effect $E[Y|do(X)]$ with continuous target $Y$ and treatments $X$ under selection bias and confounding when having access to external data for $X$ and $Z$ not underlying the selection mechanism. We derived conditions (Assumption 2.3) under which the causal effect is identifiable and s-recoverable (Theorem 2.4). Assuming linearity with Gaussian errors, we proposed a generalized estimator, the Two-Step Regression Estimator (TSR), in line with our theoretical results. We discussed how TSR simplifies in different situations, e.g., when confounding is absent, and how to introduce non-linearity. For a minimal example with uncorrelated $X$ and proxies $Z$, we proved that the variance of TSR is at most of the same magnitude as of repeated regression (RR) [Boeken et al., 2023], and confirmed this result through simulation studies. Further, we validated our estimator through extensive simulation studies. It became evident that an estimator capable of handling both selection bias and confounding is necessary because in wide ranges of the support of $X$, even the centralized 95%-area of the estimates for $E[Y|X]$ did not cover the underlying causal effect $E[Y|do(X)]$. Last, we found that adding a ridge penalty to OLS when applying TSR and RR can result in a lower variance of the causal effect estimates, but introduces a bias for some examples.

**Limitations and Future Work** Although our estimator covers a range of different settings, we need to assume access to proxy variables. In Appendix B.4, we provide an experiment to evaluate slight violations of this assumption. Another important assumption is access to external unbiased data for $X$ and $Z$. This assumption may hold when the label is costly but the covariates can be accessed through other databases. As in the loan default example (Figure 1), information about job type, income, etc., may be accessible in other databases which do not contain measurements of the loan default. In other scenarios, however, this assumption may be restrictive (Appendix A.4). However, for TSR, depending on the specific setting, observing $X$, $Z^-$ and $Z^+$ separately unbiased may be sufficient. They only need to be observed jointly if variables appear together within an expectation term - such as $X$ and $Z^-$ in $E[Z^- \mid X]$. Hence, the conditions for TSR are more attainable in practical applications. In contrast, computing $\widetilde{Y}$ for RR requires access to data containing $X$ and $Z$ jointly.

For future work, we plan to relax some of our assumptions, and work on more flexible estimators that can, e.g., be instantiated through neural networks, and study more assumption violations. For instance the effect of missing variables, or absence of Gaussian errors.

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

# Appendix

**Table of Contents**

# A Theory

In this section, the related work is presented in more detail, although it has already been addressed earlier in the paper. After the related work section, we will give the definition of the do-calculus, which is fundamental for our theoretic results and the definition of s-recoverability, we referred to stating the definition of s-recoverability with access to external data. After this, we state and discuss the selection backdoor criterion by Bareinboim et al. [2014] and the generalized adjustment criterion type 3 by Correa et al. [2018].

## A.1 Related Work

One origin of selection bias is systematic missingness or preferential inclusion of datapoints, which is a well-studied problem. A comprehensive overview of handling missing data is given by Little and Rubin [2002]. Fundamental achievements of research about missing data settings are given in Dempster et al. [1977] on the EM algorithm, Heckman [1979] on correcting for selection bias in linear regression and Rosenbaum and Rubin [1984] on bias reduction through subclassification on the propensity score. Another important aspect is to actually detect if observed data is subject to selection bias, which is a topic that has been investigated by Daniel et al. [2012] and Kaltenpoth and Vreeken [2023].

Independent of whenever we know through detection methods or domain experts that a dataset is affected by selection bias, it is necessary to properly correct rather than ignore selection bias. This has been emphasized by various authors, for instance by Sharma et al. [2022] and Castro et al. [2020]. Several approaches have been proposed to address selection bias. Examples are Mohan and Pearl [2021] who derived a consistent estimation method in missing data problems and Goel et al. [2021] who investigated fairness algorithms. A data-driven variable decomposition ($D^2VD$) that jointly optimizes separation of variables into confounders and adjustment variables to handle confounding but not selection bias is proposed by Kuang et al. [2017], where the focus lies on the estimation of the average treatment effect from high dimensional data from observational studies. Further, Liu et al. [2024] employed proxy-based two-stage generalized linear regression models (GLMs) to adjust for unmeasured confounding in unbiased data.

Extensive work has been done on conditions that ensure the causal effect to occur identifiable and s-recoverable. Bareinboim and Pearl [2012] derived a complete condition indicating feasibility of recoverability of the odds ratio (OR) from selection biased data and offered a method enabling to recover other effect measures than OR from selection bias using instrumental variables. Pearl [2012] and Bareinboim et al. [2014] considered the fundamental problem of the identifiability of $P(Y \mid X)$ based on data potentially underlying a selection bias. Bareinboim et al. [2014] in detail discussed the concept of s-recoverability and expanded it to cases that are only s-recoverable under access to additional external data not underlying the selection mechanism. They defined assumptions under which $P(Y \mid X)$ or $P(Y \mid do(X))$ can be ensured to be s-recoverable having access to external data. Forré and Mooij [2020] extended the backdoor and selection backdoor criterion to a general class of structural causal models allowing for cycles, and Chen et al. [2024] introduced a conditioning operation on structural causal models allowing to model selection bias in a principled manner akin to confounding. The results of Bareinboim et al. [2014] were extended to the topic of data fusion [Bareinboim and Pearl, 2016] by assuming access to multiple datasets, where some of them may be affected by selection bias. Further, Correa and Bareinboim [2017] established complete conditions in absence of external data and for the setting in which all proxy variables are observed externally, which has been extended to cover the case in which a subset of the proxy variables is observed externally [Correa et al., 2018]. Our identifiably result covers a different setting, e.g., the graph shown in Figure 1, as we discuss in more detail in Appendix A.4. Similar results that - in contrast to our work - require access to experimental data are reported in Singh and Zhou [2022]. Colnet et al. [2024] devise an approach that requires selected and external data. However, they focus on binary treatments to estimate the average treatment effect. Tchetgen Tchetgen et al. [2024] and Miao et al. [2018] address unmeasured confounding through the use of proxy variables, which aligns with certain aspects of our adjustment and Louizos et al. [2017] build on VAEs to adjust for confounding with similar assumptions.

Most closely related to our approach is the work by Boeken et al. [2023], who set their focus on proposing practical estimators to recover $E[Y \mid X]$ from selection-biased data with proxy

variables. They proposed several estimators from which, repeated regression had the most promising performance. Under unconfoundedness, $E[Y \mid do(X)] = E[Y \mid X]$. Hence, the repeated regression from Boeken et al. [2023] is applicable for the causal effect estimation in absence of confounding. Here, we extend their setting but focus on recovering causal effects from selection-biased data, where we, to some extend, allow for (unobserved) confounding. We derive criteria for identifiability and s-recoverability of the causal effect and propose an empirical estimator for this case.

## A.2 Do-Calculus

For completeness, we restate the rules of do-calculus, which we need to derive our theoretical results below [Pearl, 2009, Chapter 3].

**Definition A.1** (rules of do-calculus). For arbitrary disjoint sets of nodes $X, Y, Z$, and $W$ in a causal DAG $\mathcal{G}$, we denote the graph obtained by deleting all edges pointing towards a node in $X$ by $\mathcal{G}_{\overline{X}}$. Similarly, the graph obtained by deleting all edges pointing away from a node in $X$ by $\mathcal{G}_{\underline{X}}$. The graph obtained by deleting edges pointing towards nodes in $X$ and edges pointing away from nodes in $Z$ is denoted by $\mathcal{G}_{\overline{X}, \underline{Z}}$.

1. (Insertion / deletion of observations):
   $P(Y = y \mid do(X = x), Z = z, W = w) = P(Y = y \mid do(X = x), W = w)$
   if $(Y \perp\!\!\!\perp_{\mathcal{G}_{\overline{X}}} Z \mid \{X, W\})$

2. (Action / observation exchange):
   $P(Y = y \mid do(X = x), do(Z = z), W = w) = P(Y = y \mid do(X = x), Z = z, W = w)$
   if $Y \perp\!\!\!\perp_{\mathcal{G}_{\overline{X}, \underline{Z}}} Z \mid \{X, W\})$

3. (Insertion / deletion of actions):
   $P(Y = y \mid do(X = x), do(Z = z), W = w) = P(Y = y \mid do(X = x), W = w)$
   if $(Y \perp\!\!\!\perp_{\mathcal{G}_{\overline{X}, \overline{Z(W)}}} Z \mid \{X, W\})$,
   where $Z(W)$ is the set of nodes in $Z$ not being ancestors of any node in $W$ in $\mathcal{G}_{\overline{X}}$

## A.3 Recoverability

Bareinboim et al. [2014] defined s-recoverability as follows:

**Definition A.2** (s-recoverability). Given a causal DAG model $(\mathcal{G}, P)$ augmented with a node $S$, the distribution $Q = P(Y \mid X)$ is said to be s-recoverable from selection biased data in $\mathcal{G}_s$ if the assumptions embedded in the causal model renders $Q$ expressible in terms of the distribution $P(V \mid S = 1)$ under selection bias. Formally, for every two probability distributions $P_1$ and $P_2$ compatible with $\mathcal{G}_s$, $P_1(V = v \mid S = 1) = P_2(V = v \mid S = 1) > 0$ implies $P_1(Y = y \mid X = x) = P_2(Y = y \mid X = x)$.

## A.4 Discussion of Assumptions

In the following, we compare our assumptions to prior work, where we first review the selection backdoor criterion proposed by Bareinboim et al. [2014], as well as the repeated regression estimator by Boeken et al. [2023], and then discuss the generalized adjustment criterion derived by Correa and Bareinboim [2017].

**Selection backdoor criterion** Bareinboim et al. [2014] proposed assumptions, given in Assumption A.3, under which the causal effect $P(Y \mid do(X))$ is, as stated in Theorem A.4, identifiable and s-recoverable.

**Assumption A.3** (Selection backdoor criterion [Bareinboim et al., 2014]). The variables $Z$ can be decomposed as $Z = Z^+ \cup Z^-$, where $Z^+$ are non-descendants of $X$ and $Z^-$ are descendants of $X$.

1. $X$ and $Z$ block all paths between $S$ and $Y$, namely $Y \perp\!\!\!\perp_{\mathcal{G}} S \mid \{X, Z\}$

2. $Z^+$ blocks all backdoor paths from $X$ to $Y$, namely $(X \perp\!\!\!\perp_{\mathcal{G}_{\underline{X}}} Y \mid Z^+)$

3. $X$ and $Z^+$ block all paths between $Z^-$ and $Y$, namely $Z^- \perp\!\!\!\perp_{\mathcal{G}} Y \mid \{X, Z^+\}$

553   4. $Z \cup \{X, Y\} \subset M$, where variables $M$ are collected under selection bias (dataset $\mathcal{S}$) and $Z \subset T$,
554      where $T$ is collected in the population-level (dataset $\mathcal{D}$).

555 In comparison to the assumption above, in Assumption 2.3, we do not need (3.) which characterizes
556 the relationship of $Z^+$ and $Z^-$, whereas we maintain the first and second subpoint. On the other
557 hand, we need to observe $X$ also in the sample $\mathcal{D}$, not underlying the selection bias if $Z^- \neq \emptyset$.

558 **Theorem A.4** (Selection backdoor adjustment [Bareinboim et al., 2014])**.** *If $Z$ satisfies the selec-*
559 *tion backdoor criterion (Assumption A.3) relative to $(X, Y)$ and $(M, T)$, then the causal effect*
560 $P(Y \mid do(X))$ *is identifiable, s-recoverable and can be expressed as*

$$P(Y = y \mid do(X)) = \int_z P(Y = y \mid X, Z = z, S = 1)P(Z = z)dz \ .$$

561 Further, recall that for RR we need to assume PMAR as well as that $X$ and $Y$ are not confounded
562 when aiming to estimate $E[Y \mid do(X)]$. We want to explain the relationship between the assumptions
563 for RR, Assumption A.3 proposed by Bareinboim et al. [2014], and our Assumption 2.3 for TSR
564 based on the four DAGs in Figure 2. First, note that Assumption 2.3 is met for all of the four cases.
565 The setting in Figure 2 (a) is met by all of the three assumptions. In contrast, in the setting in Figure 2
566 (b) Assumption A.3 (3.) is violated as the edge between $Z^-$ and $Y$ can not be blocked by $X$ and
567 $Z^+$. For the setting in Figure 2 (c) it is exactly the opposite. Here, RR does not recover the causal
568 effect but only $E[Y|X]$, whereas Assumption A.3 is fulfilled. The setting in Figure 2 (d) violates
569 Assumption A.3 (3.), and induces confounding, which is why RR does not recover the causal effect.

570 In Section 2.3, we mentioned that Assumption 2.3 requires $X$ and $Z$ to be observed unbiased, whereas
571 for Assumption A.3 only $Z$ must be observed unbiased. That is, if $X$ is not observable unbiased,
572 and $Z^- \neq \emptyset$, there might be cases which meet Assumption A.3, but not our Assumption 2.3. It is to
573 mention, that settings could occur in which our assumption is met, but the selection backdoor criterion
574 is not met due to a lack of access to unbiased data. Think of cases for which it is difficult to observe
575 $Z^+$ unbiased, whereas observing $X$ unbiased is unproblematic. In those cases, it may be that our
576 assumption is fulfilled but the selection backdoor criterion is not. As discussed above, for example,
577 Figure 2 (b) does not satisfy the third point of Assumption A.3, whereas Assumption 2.3 is met. Swap
578 the direct path $Z^- \rightarrow Y$ to one that goes via confounder $Z^+$, $Z^- \leftarrow Z^+ \rightarrow Y$. Assuming, that $Z^+$
579 can not be observed unbiased, Assumption A.3 can not be satisfied, whereas, taking advantage of
580 Corollary 2.7, Assumption 2.3 can be satisfied if $(X, Z^-)$ are observable unbiased.

581 **Generalized Adjustment Criterion 3 (GACT3)**   Correa et al. [2018] proposed assumptions,
582 given in Definition A.8, under which the causal effect $P(Y|do(X))$ is, as stated in Theorem A.10,
583 identifiable and s-recoverable, which requires some preliminary definitions by Correa et al. [2018],
584 which we state below.

585 **Definition A.5** (Proper Causal Path)**.** Let $X$ and $Y$ be sets of nodes. A causal path from a node in $X$
586 to a node in $Y$ is called proper if it does not intersect $X$ except at the starting point.

587 **Definition A.6** (Proper Backdoor Graph)**.** Given a causal DAG model $(\mathcal{G}, P)$ and disjoint subsets $X$
588 and $Y$ of variables. The proper backdoor graph, denoted as $\mathcal{G}_{XY}^{pbd}$, is obtained from $\mathcal{G}$ by removing
589 the first edge of every proper causal path from $X$ to $Y$.

590 **Definition A.7** (Adjustment Pair)**.** Given a causal DAG model $(\mathcal{G}, P)$ augmented with a node $S$,
591 disjoint sets of variables $X, Y, Z$, and a set $Z^T \subset Z$, $(Z, Z^T)$ is said to be an adjustment pair for
592 recovering the causal effect of $X$ on $Y$ if for every model compatible with $\mathcal{G}$, $P(Y{=}y \mid do(X{=}x))$
593 can be expressed as

$$\sum_z P(Y = y \mid X = x, Z = z, S = 1)P(Z = z \setminus Z^T = z^T \mid Z^T = z^T, S = 1)P(Z^T = z^T) \ .$$

594

595 **Assumption A.8** (Generalized Adjustment Criterion Type 3 (GACT3))**.** Given a causal DAG model
596 $(\mathcal{G}, P)$ augmented with a node $S$, disjoint sets of variables $X, Y, Z$ and set $Z^T \subset Z$; $(Z, Z^T)$ is an
597 admissible pair relative to $X, Y$ in $\mathcal{G}$ if:

598   1. No element in $Z$ is a descendant in $\mathcal{G}_{\overline{X}}$ of any $W \notin X$ lying on a proper causal path from $X$ to $Y$.

599   2. All non-causal paths in $\mathcal{G}$ from $X$ to $Y$ are blocked by $Z$ and $S$.

3. $Z^T$ d-separates $Y$ from $S$ in the proper backdoor graph, i.e. $(Y \perp\!\!\!\perp S \mid Z^T)_{\mathcal{G}_{XY}^{pbd}}$

**Theorem A.9** (Admissible Pairs are Adjustment Pairs). *$Z, Z^T$ is an adjustment pair for $X$, $Y$ in $\mathcal{G}$ if and only if it is admissible by Assumption A.8.*

**Corollary A.10** (Causal Effects Recovery by Adjustment). *Given a causal DAG model $(\mathcal{G}, P)$ augmented with a node $S$ representing the selection mechanism. Let $V$ be the set of variables measured under selection bias, and $T \subset V$ the set of variables measured externally in the overall population. Consider disjoint sets of variables $X, Y \subset V$, then the causal effect $P(Y = y \mid do(X = x))$ is recoverable from $\{P(V = v \mid S = 1), P(T = t)\}$ by the adjustment expression in Definition A.7 while $Z^T \subset T$, in every model inducing $\mathcal{G}$ if and only if $(Z, Z^T)$ is an admissible pair relative to $X$, $Y$ in $\mathcal{G}$ according to Assumption A.8.*

The assumptions in Assumption A.8 are met in Figure 2 (a)–(c), but are not admissible for (d) in cases where $U$ is not contained in $Z$. Such cases could occur if either we did not include $U$ in $Z$, or it is unobserved. In either case, not all non-causal paths from $X$ to $Y$ can be blocked by $Z$ and $S$. Conditioning on $Z^-$ opens the path $X \to Z^- \leftarrow U \to Y$. However, $Z^-$ must be included into $Z$ to meet the PMAR assumption.

## A.5  Proofs

In this section, we provide the proofs.

**Theorem 2.4.** *Under Assumption 2.3, the causal effect $E[Y \mid do(X)]$ is identifiable, s-recoverable and can be expressed as follows*

$$\int_{z^+} E[E[Y \mid X, Z^+ = z^+, Z^-, S = 1] \mid X, Z^+ = z^+]P(Z^+ = z^+)dz^+ .$$

*Proof.* We can express the expected causal effect $E[Y \mid do(X)]$ as

$$E[Y \mid do(X)] = \int_{z^+} \underbrace{E[Y \mid do(X), Z^+ = z^+]}_{\substack{= E[Y|X,Z^+=z^+] \\ Y \perp\!\!\!\perp_{\mathcal{G}_{\underline{X}}} X | Z^+}} \underbrace{P(Z^+ = z^+ \mid do(X))}_{\substack{= P(Z^+=z^+) \\ \text{non-desc}}} dz^+$$

$$= \int_{z^+} \underbrace{E[Y \mid X, Z^+ = z^+]}_{=E[E[Y|X,Z^+=z^+,Z^-]|X,Z^+=z^+]} P(Z^+ = z^+)dz^+$$

$$= \int_{z^+} E[\underbrace{E[Y \mid X, Z^+ = z^+, Z^-]}_{=E[Y|X,Z^+=z^+,Z^-,S=1]} \mid X, Z^+ = z^+]P(Z^+ = z^+)dz^+$$

$$= \int_{z^+} E[E[Y \mid X, Z^+ = z^+, Z^-, S = 1] \mid X, Z^+ = z^+]P(Z^+ = z^+)dz^+$$

The first and third row follow from the law of total expectation. In row two, we can apply the second rule of do-calculus (cf. Definition A.1 in Appendix A.2) since $Y \perp\!\!\!\perp_{\mathcal{G}_{\underline{X}}} X \mid Z^+$ (Assumption 2.3 (2.)), and $Z^+$ is non-descendant of $X$. Assumption 2.3 (1.) ensures the final equality.

Following the proof of s-recoverability by Bareinboim et al. [2014] for Theorem A.4, as the causal effect can be represented in probability terms of the selected sample and of the external data, along with Assumption 2.3, the achieved expression ensures s-recoverability.

□

**Theorem 2.6.** *Under Assumption 2.3 and Assumption 2.5, the causal effect $E[Y \mid do(X)]$ is identifiable, s-recoverable and can be expressed as*

$$E[Y \mid do(X = x)] = \beta_0 + \beta_1 x + \beta_2 E[Z^+] + \beta_3 \int_{z^+} E[Z^- \mid X = x, Z^+ = z^+]P(Z^+ = z^+)dz^+.$$

(1)

*Proof.*

$$E[Y|do(X = x)] = \int_{z^+} E[E[Y \mid X = x, Z^+ = z^+, Z^-, S = 1] \mid X = x, Z^+ = z^+]P(Z^+ = z^+)dz^+$$

$$= \int_{z^+} E[\beta_0 + \beta_1 X + \beta_2 Z^+ + \beta_3 Z^- \mid X = x, Z^+ = z^+]P(Z^+ = z^+)dz^+$$

$$= \int_{z^+} (\beta_0 + \beta_1 x + \beta_2 z^+ + \beta_3 E[Z^- \mid X = x, Z^+ = z^+])P(Z^+ = z^+)dz^+$$

$$= \beta_0 + \beta_1 x + \beta_2 \int_{z^+} z^+ P(Z^+ = z^+)dz^+$$

$$+ \beta_3 \int_{z^+} E[Z^- \mid X = x, Z^+ = z^+]P(Z^+ = z^+)dz^+$$

$$= \beta_0 + \beta_1 x + \beta_2 E[Z^+] + \beta_3 \int_{z^+} E[Z^- \mid X = x, Z^+ = z^+]P(Z^+ = z^+)dz^+$$

$\square$

**Corollary 2.7.** *If $Z^+$ blocks all backdoor paths between $X$ and $Z^-$, the integral from above reduces to $E[Z^- \mid do(X)]$. If additionally, $X$ and $Z^-$ are not confounded, it reduces to $E[Z^- \mid X]$.*

*Proof.*

$$E[Z^- \mid do(X)] = \int_{z^+} E[Z^- \mid do(X), Z^+ = z^+]P(Z^+ = z^+ \mid do(X))dz^+$$

$$= \int_{z^+} E[Z^- \mid X, Z^+ = z^+]P(Z^+ = z^+)dz^+$$

In the derivation above, we exploit that $Z^- \perp\!\!\!\perp X \mid Z^+$ in $\mathcal{G}_{\underline{X}}$ along with the second rule of do-calculus, the Law of total expectation and $Z^+$ being non-descendant of $X$. Further, $E[Z^- \mid X] = E[Z^- \mid do(X)]$, if $X$ and $Z^-$ are not confounded. $\square$

## A.6 When do RR and TSR coincide?

RR and TSR coincide when $\hat{\mu}_{TSR}(x) = \hat{\beta}_0 + \hat{\beta}_1 x + \hat{\beta}_2 \widehat{E}[Z^- \mid X = x]$, which is the case in absence of confounding and $Z^+ = \emptyset$ (implying $Z = Z^-$) as in Figure 2 (b). As shown below, this result can be expanded to the nonlinear case as described in Section 3.2.

The first step of regression for both estimators, yields

$$\hat{\beta} = (\hat{\beta}_x, \hat{\beta}_z) \,,$$

where $\hat{\beta}_x := (\hat{\beta}_0, \hat{\beta}_1)$ and $\hat{\beta}_z := \hat{\beta}_2$.

Then TSR is given by

$$\hat{\mu}_{TSR}(x) = \hat{\beta}_x \overrightarrow{x} + \hat{\beta}_z ((B_X^T B_X)^{-1} B_X^T B_Z)^T \overrightarrow{x} \,,$$

where $\overrightarrow{x} := (1, x^T)^T$, $B_X := (1, X_i^T)_{i \in \mathcal{D}} \in \mathbb{R}^{|\mathcal{D}| \times (p+1)}$ and $B_Z := (Z_i^T)_{i \in \mathcal{D}} \in \mathbb{R}^{|\mathcal{D}| \times d}$.

RR is derived as follows:

$$\hat{\mu}_{RR}(x) = (\overrightarrow{x})^T (B_X^T B_X)^{-1} B_X^T (B_X (\hat{\beta}_x)^T + B_Z (\hat{\beta}_z)^T)$$

$$= (\overrightarrow{x})^T (B_X^T B_X)^{-1} B_X^T B_X (\hat{\beta}_x)^T + (\overrightarrow{x})^T (B_X^T B_X)^{-1} B_X^T B_Z (\hat{\beta}_z)^T$$

$$= (\overrightarrow{x})^T (\hat{\beta}_x)^T + (\overrightarrow{x})^T (B_X^T B_X)^{-1} B_X^T B_Z (\hat{\beta}_z)^T$$

$$= \hat{\beta}_x \overrightarrow{x} + \hat{\beta}_z ((B_X^T B_X)^{-1} B_X^T B_Z)^T \overrightarrow{x} \,.$$

650 This verifies the equality $\hat{\mu}_{RR}(x) = \hat{\mu}_{TSR}(x)$ for the linear case. With the same calculation but other
651 definitions for $B_X, B_Z, \hat{\beta}_x, \hat{\beta}_z$ and $\overrightarrow{x}$, the setting can be generalized - for example to OLS with
652 impacts up to a specific degree.

653

654 In settings, for which we cannot exclude confounding and $Z^- = \emptyset$ (implying $Z = Z^+$),
655 as in Figure 2 (c), TSR, given by $\hat{\mu}_{TSR}(x) = \hat{\beta}_0 + \hat{\beta}_1 x + \hat{\beta}_2 \widehat{E}[Z^+]$ in this setting, does not exactly
656 equal the RR estimation. We consider the case, where $X$ and $Z$ are of dimension one.

657 As we will show below, using OLS,

$$\hat{\mu}_{TSR}(x) = \hat{\beta}_0 + \hat{\beta}_1 x + \hat{\beta}_2 \overline{Z^+}$$

658 and

$$\hat{\mu}_{RR}(x) = \hat{\mu}_{TSR}(x) + \hat{\beta}_2 \frac{\widehat{Cov}[Z^+, X]}{\widehat{Var}[X]}(x - \bar{X}) \ .$$

659 The derivation is at follows. First, define $\widetilde{Y}$ for the second regression step for RR as

$$\widetilde{Y} := \hat{\beta}_0 + \hat{\beta}_1 X + \hat{\beta}_2 Z^+ \ .$$

660 Then, running again a simple linear regression, we search for estimators $\hat{\alpha}_0$ as intercept and $\hat{\alpha}_1$ as
661 coefficient of $X$ for the second step of RR. They are given as follows:

$$\hat{\alpha}_1 = \frac{\widehat{Cov}[\widetilde{Y}, X]}{\widehat{Var}[X]} = \hat{\beta}_1 + \hat{\beta}_2 \frac{\widehat{Cov}[Z^+, X]}{\widehat{Var}[X]}$$

662

$$\begin{aligned}
\hat{\alpha}_0 &= \bar{\widetilde{Y}} - \hat{\alpha}_1 \bar{X} \\
&= \hat{\beta}_0 + \hat{\beta}_1 \bar{X} + \hat{\beta}_2 \overline{Z^+} - \hat{\alpha}_1 \bar{X} \\
&= \hat{\beta}_0 + \hat{\beta}_1 \bar{X} + \hat{\beta}_2 \overline{Z^+} - \left( \hat{\beta}_1 + \hat{\beta}_2 \frac{\widehat{Cov}[Z^+, X]}{\widehat{Var}[X]} \right) \bar{X} \\
&= \hat{\beta}_0 + \hat{\beta}_2 \overline{Z^+} - \hat{\beta}_2 \frac{\widehat{Cov}[Z^+, X]}{\widehat{Var}[X]} \bar{X} \ .
\end{aligned}$$

663 This results into

$$\begin{aligned}
\hat{\mu}_{RR}(x) &= \hat{\alpha}_0 + \hat{\alpha}_1 x \\
&= \hat{\beta}_0 + \hat{\beta}_2 \overline{Z^+} - \hat{\beta}_2 \frac{\widehat{Cov}[Z^+, X]}{\widehat{Var}[X]} \bar{X} + \left( \hat{\beta}_1 + \hat{\beta}_2 \frac{\widehat{Cov}[Z^+, X]}{\widehat{Var}[X]} \right) x \\
&= \underbrace{\hat{\beta}_0 + \hat{\beta}_1 x + \hat{\beta}_2 \overline{Z^+}}_{= \hat{\mu}_{TSR}(x)} + \hat{\beta}_2 \frac{\widehat{Cov}[Z^+, X]}{\widehat{Var}[X]}(x - \bar{X}) \ .
\end{aligned}$$

664 Even if $Cov[Z^+, X] = 0$, $\widehat{Cov}[Z^+, X] \neq 0$ for finite samples.

665

666 With $\mathcal{S} \cap \mathcal{D} = \emptyset$,

$$E\left[ \hat{\beta}_2 \frac{\widehat{Cov}[Z^+, X]}{\widehat{Var}[X]}(x - \bar{X}) \right] = E[\hat{\beta}_2] E\left[ \frac{\widehat{Cov}[Z^+, X]}{\widehat{Var}[X]}(x - \bar{X}) \right] \ ,$$

667 which diminishes for $n \to \infty$, provided that the estimate of the variance in the denominator does not
668 asymptotically approach zero.

### A.6.1 Bias and Variance

670 For the calculation of the bias and variance of RR, we restrict ourselves to the case, where $X$ and $Z$
671 are of dimension one. We make use of the explicit form $\hat{\mu}_{RR}(x) = \hat{\beta}_0 + \hat{\beta}_1 x + \hat{\beta}_2 (\hat{\alpha}_0 + \hat{\alpha}_1 x)$ and
672 justify this expression in the following, deriving the repeated regression estimator by hand. For the

673 second regression, we estimate a simple linear regression and therefore set $\widehat{E}[\widetilde{Y} \mid X = x] = \hat{\delta}_0 + \hat{\delta}_1 x$.

674 The derivation of $\hat{\delta}_0$ and $\hat{\delta}_1$ will be given in the following:

675 First, we use that for the simple linear regression using OLS, $\hat{\delta}_1 = \frac{\widehat{Cov}[\widetilde{Y}, X]}{\widehat{Var}[X]}$, such that

$$\widehat{Var}[X]\hat{\delta}_1 = \widehat{Cov}[\widetilde{Y}, X] = \widehat{Cov}[\hat{\beta}_0 + \hat{\beta}_1 X + \hat{\beta}_2 Z, X] = \underbrace{\widehat{Cov}[\hat{\beta}_0, X]}_{=0} + \underbrace{\widehat{Cov}[\hat{\beta}_1 X, X]}_{\hat{\beta}_1 \widehat{Var}[X]} + \underbrace{\widehat{Cov}[\hat{\beta}_2 Z, X]}_{=\hat{\beta}_2 \widehat{Cov}[Z,X]} .$$

676 Consequently, we arrive at

$$\hat{\delta}_1 = \hat{\beta}_1 + \hat{\beta}_2 \frac{\widehat{Cov}[Z, X]}{\widehat{Var}[X]} .$$

677 From this, we can calculate $\hat{\delta}_0$ as follows:

$$\hat{\delta}_0 = \bar{\widetilde{Y}} - \hat{\delta}_1 \bar{X} \underset{\bar{\widetilde{Y}} = \hat{\beta}_0 + \hat{\beta}_1 \bar{X} + \hat{\beta}_2 \bar{Z}}{=} \hat{\beta}_0 + \hat{\beta}_1 \bar{X} + \hat{\beta}_2 \bar{Z} - \hat{\beta}_1 \bar{X} - \hat{\beta}_2 \frac{\widehat{Cov}[Z, X]}{\widehat{Var}[X]} \bar{X} = \hat{\beta}_0 + \hat{\beta}_2 \bar{Z} - \hat{\beta}_2 \frac{\widehat{Cov}[Z, X]}{\widehat{Var}[X]} \bar{X} .$$

678 Plugging in $\hat{\delta}_0$ and $\hat{\delta}_1$, the repeated regression estimator can be expressed as

$$\begin{aligned}
\hat{\mu}_{RR}(x) &= \hat{\delta}_0 + \hat{\delta}_1 x \\
&= \hat{\beta}_0 + \hat{\beta}_2 \bar{Z} - \hat{\beta}_2 \frac{\widehat{Cov}[Z, X]}{\widehat{Var}[X]} \bar{X} + \hat{\beta}_1 x + \hat{\beta}_2 \frac{\widehat{Cov}[Z, X]}{\widehat{Var}[X]} x \\
&= \hat{\beta}_0 + \hat{\beta}_1 x + \hat{\beta}_2 \bigg( \underbrace{\bar{Z} - \frac{\widehat{Cov}[X, Z]}{\widehat{Var}[X]} \bar{X}}_{=\hat{\alpha}_0} + \underbrace{\frac{\widehat{Cov}[X, Z]}{\widehat{Var}[X]} x}_{=\hat{\alpha}_1} \bigg) \\
&= \hat{\beta}_0 + \hat{\beta}_1 X + \hat{\beta}_2 (\hat{\alpha}_0 + \hat{\alpha}_1 x) ,
\end{aligned}$$

679 where $\hat{\alpha}_0$ and $\hat{\alpha}_1$ denote the OLS coefficient estimates of $\alpha_0$ and $\alpha_1$ corresponding to the simple
680 linear regession model $E[Z|X = x] = \alpha_0 + \alpha_1 x$.

681

682 **Unbiasedness** Now, we can calculate the empirical mean and discuss its bias for RR and TSR:

$$\begin{aligned}
E[\hat{\mu}_{RR}(x)] &= E[\hat{\beta}_0 + \hat{\beta}_1 x + \hat{\beta}_2 (\hat{\alpha}_0 + \hat{\alpha}_1 x)] \\
&\underset{\text{linearity}}{=} E[\hat{\beta}_0] + E[\hat{\beta}_1]x + E[\hat{\beta}_2 (\hat{\alpha}_0 + \hat{\alpha}_1 x)] \\
&= E[\hat{\beta}_0] + E[\hat{\beta}_1]x + E[\hat{\beta}_2]E[\hat{\alpha}_0 + \hat{\alpha}_1 x] + Cov[\hat{\beta}_2, \hat{\alpha}_0 + \hat{\alpha}_1 x] \\
&\underset{\text{first step model correctly specified}}{=} \beta_0 + \beta_1 x + \beta_2 E[\hat{\alpha}_0 + \hat{\alpha}_1 x] + Cov[\hat{\beta}_2, \hat{\alpha}_0 + \hat{\alpha}_1 x] \\
&\underset{\text{second step model correctly specified}}{=} \beta_0 + \beta_1 x + \beta_2 (\alpha_0 + \alpha_1 x) + Cov[\hat{\beta}_2, \hat{\alpha}_0 + \hat{\alpha}_1 x] \\
&\underset{\alpha_0 = E[\overline{Z^+}] = E[Z^+], \ \alpha_1 = 0}{=} \beta_0 + \beta_1 x + \beta_2 E[Z^+] + \underbrace{Cov[\hat{\beta}_2, \hat{\alpha}_0 + \hat{\alpha}_1 x]}_{= \begin{cases} \neq 0 & \mathcal{S} \subset \mathcal{D} \\ = 0 & \mathcal{S} \cap \mathcal{D} = \emptyset \end{cases}}
\end{aligned}$$

$$\begin{aligned}
E[\hat{\mu}_{TSR}(x)] &= E[\hat{\beta}_0 + \hat{\beta}_1 x + \hat{\beta}_2 \overline{Z^+}] \\
&\overset{linearity}{=} E[\hat{\beta}_0] + E[\hat{\beta}_1] x + E[\hat{\beta}_2 \overline{Z^+}] \\
&= E[\hat{\beta}_0] + E[\hat{\beta}_1] x + E[\hat{\beta}_2] E[\overline{Z^+}] + Cov[\hat{\beta}_2, \overline{Z^+}] \\
&\overset{\text{first step model correctly specified}}{=} \beta_0 + \beta_1 x + \beta_2 E[\overline{Z^+}] + Cov[\hat{\beta}_2, \overline{Z^+}] \\
&= \beta_0 + \beta_1 x + \beta_2 E[Z^+] + \underbrace{Cov[\hat{\beta}_2, \overline{Z^+}]}_{= \begin{cases} \neq 0 & \mathcal{S} \subset \mathcal{D} \\ = 0 & \mathcal{S} \cap \mathcal{D} = \emptyset \end{cases}}
\end{aligned}$$

683 We discuss the bias for the general case for TSR, where $X$ and $Z$ are not restricted to dimension one.
684 Suppose we are in the setting with

$$\hat{\mu}_{TSR}(x) = \hat{\beta}_0 + \hat{\beta}_1 x + \hat{\beta}_2 \overline{Z^+} + \hat{\beta}_3(\hat{\gamma}_0 + \hat{\gamma}_1 x + \hat{\gamma}_2 \overline{Z^+}) \ .$$

685 Due to the unbiasedness of OLS and $\mathcal{S} \cap \mathcal{D} = \emptyset$, the bias of TSR reduces to the bias of just $\hat{\beta}_3 \hat{\gamma}_2 \overline{Z^+}$
686 w.r.t. $\beta_3 \gamma_2 E[Z^+]$, which is given by

$$\begin{aligned}
\text{Bias}[\hat{\mu}_{TSR}] &= E[\hat{\beta}_3 \hat{\gamma}_2 \overline{Z^+}] - \beta_3 \gamma_2 E[Z^+] \\
&= \underbrace{E[\hat{\beta}_3]}_{=\beta_3} E[\hat{\gamma}_2 \overline{Z^+}] - \beta_3 \gamma_2 E[Z^+] \\
&= \beta_3 (E[\hat{\gamma}_2 \overline{Z^+}] - \gamma_2 E[Z^+]) \\
&= \beta_3 (E[\hat{\gamma}_2 \overline{Z^+}] - E[\hat{\gamma}_2] E[\overline{Z^+}]) \ ,
\end{aligned}$$

687 where for the second row we used $\mathcal{S} \cap \mathcal{D} = \emptyset$ and unbiasedness of OLS, and for the fourth we used
688 unbiasedness of OLS and linearity of the expectation together with the fact that the repetitions of $Z^+$
689 are identically distributed. We can ensure unbiasedness by splitting the unbiased dataset into two
690 disjoint subsets - one to estimate $\hat{\gamma}_0, \hat{\gamma}_1$ and $\hat{\gamma}_2$ via the second regression, and one to compute the
691 empirical mean of $Z^+$ - although this comes at the cost of reduced efficiency.
692

693 Next, we proof the result comparing the variances of RR and TSR, which is restated below.

694 **Theorem 3.2.** *Under Assumption 3.1 and $\mathcal{S} \cap \mathcal{D} = \emptyset$, let $\widehat{E}[Z^+ \mid X = x] = \hat{\alpha}_0 + \hat{\alpha}_1 x$ be the OLS*
695 *estimator of the second step for $\hat{\mu}_{RR}(x)$ and $\bar{X} = \frac{1}{|\mathcal{D}|} \sum_{i=1}^{|\mathcal{D}|} X_i$, then*

$$Var[\hat{\mu}_{RR}(x)] - Var[\hat{\mu}_{TSR}(x)] = Var[\hat{\beta}_2(\hat{\alpha}_1(x - \bar{X}))] \geq 0 \ .$$

696 *Proof.* First, we derive the variance for $\hat{\mu}_{RR}(x)$.

$$\begin{aligned}
Var[\hat{\mu}_{RR}(x)] &= Var[\hat{\beta}_0 + \hat{\beta}_1 x + \hat{\beta}_2 (\underbrace{\underbrace{\hat{\alpha}_0}_{=\overline{Z^+} - \hat{\alpha}_1 \bar{X}} + \hat{\alpha}_1 x)}_{=\widehat{E}[Z^+|X]}] \\
&= Var[\hat{\beta}_0 + \hat{\beta}_1 x + \hat{\beta}_2 \overline{Z^+}] + Var[\hat{\beta}_2 \hat{\alpha}_1 (x - \bar{X})] + 2Cov[\hat{\beta}_0 + \hat{\beta}_1 x + \hat{\beta}_2 \overline{Z^+}, \hat{\beta}_2 \hat{\alpha}_1 (x - \bar{X})]
\end{aligned}$$

697 Based on the above result, we can write the difference in variance of both estimators as
698 $\Delta = Var[\hat{\mu}_{RR}(x)] - Var[\hat{\mu}_{TSR}(x)]$, where we can express $\Delta$ as

$$\Delta = Var[\hat{\mu}_{RR}(x)] - Var[\hat{\beta}_0 + \hat{\beta}_1 x + \hat{\beta}_2 \overline{Z^+}]$$

$$= Var[\hat{\beta}_2 \hat{\alpha}_1 (x - \bar{X})] + 2Cov[\hat{\beta}_0 + \hat{\beta}_1 x + \hat{\beta}_2 \overline{Z^+}, \hat{\beta}_2 \hat{\alpha}_1 (x - \bar{X})]$$

$$= Var[\hat{\beta}_2 \hat{\alpha}_1 (x - \bar{X})] + 2(Cov[\hat{\beta}_0, \hat{\beta}_2 \hat{\alpha}_1 x] - Cov[\hat{\beta}_0, \hat{\beta}_2 \hat{\alpha}_1 \bar{X}] + Cov[\hat{\beta}_1 x, \hat{\beta}_2 \hat{\alpha}_1 x]$$

$$- Cov[\hat{\beta}_1 x, \hat{\beta}_2 \hat{\alpha}_1 \bar{X}] + Cov[\hat{\beta}_2 \overline{Z^+}, \hat{\beta}_2 \hat{\alpha}_1 x] - Cov[\hat{\beta}_2 \overline{Z^+}, \hat{\beta}_2 \hat{\alpha}_1 \bar{X}])$$

$$= Var[\hat{\beta}_2 \hat{\alpha}_1 (x - \bar{X})] + 2 \cdot ( \quad \underbrace{E[\hat{\beta}_0 \hat{\beta}_2 \hat{\alpha}_1 x]}_{\underset{\mathcal{S} \cap \mathcal{D} = \emptyset}{=} x E[\hat{\beta}_0 \hat{\beta}_2] \underbrace{E[\hat{\alpha}_1]}_{\underset{X \perp Z^+}{=} 0}} \quad - \quad \underbrace{E[\hat{\beta}_0] E[\hat{\beta}_2 \hat{\alpha}_1 x]}_{\underset{\mathcal{S} \cap \mathcal{D} = \emptyset}{=} E[\hat{\beta}_0] x E[\hat{\beta}_2] \underbrace{E[\hat{\alpha}_1]}_{\underset{X \perp Z^+}{=} 0}}$$

$$- \quad \underbrace{E[\hat{\beta}_0 \hat{\beta}_2 \hat{\alpha}_1 \bar{X}]}_{\underset{\mathcal{S} \cap \mathcal{D} = \emptyset}{=} E[\hat{\beta}_0 \hat{\beta}_2] \underbrace{E[\hat{\alpha}_1 \bar{X}]}_{= \underbrace{E[\hat{\alpha}_1]}_{\underset{X \perp Z^+}{=} 0} E[\bar{X}]}} \quad + E[\hat{\beta}_0] \underbrace{E[\hat{\beta}_2 \hat{\alpha}_1 \bar{X}]}_{\underset{\mathcal{S} \cap \mathcal{D} = \emptyset}{=} E[\hat{\beta}_2] \underbrace{E[\hat{\alpha}_1]}_{\underset{X \perp Z^+}{=} 0} E[\bar{X}]}$$

$$+ \quad \underbrace{E[\hat{\beta}_1 x \hat{\beta}_2 \hat{\alpha}_1 x]}_{\underset{\mathcal{S} \cap \mathcal{D} = \emptyset}{=} x^2 E[\hat{\beta}_1 \hat{\beta}_2] \underbrace{E[\hat{\alpha}_1]}_{\underset{X \perp Z^+}{=} 0}} \quad - \quad \underbrace{E[\hat{\beta}_1 x] E[\hat{\beta}_2 \hat{\alpha}_1 x]}_{\underset{\mathcal{S} \cap \mathcal{D} = \emptyset}{=} E[\hat{\beta}_1] x^2 E[\hat{\beta}_2] \underbrace{E[\hat{\alpha}_1]}_{\underset{X \perp Z^+}{=} 0}}$$

$$- \quad \underbrace{E[\hat{\beta}_1 x \hat{\beta}_2 \hat{\alpha}_1 \bar{X}]}_{\underset{\mathcal{S} \cap \mathcal{D} = \emptyset}{=} x E[\hat{\beta}_1 \hat{\beta}_2] \underbrace{E[\hat{\alpha}_1 \bar{X}]}_{= \underbrace{E[\hat{\alpha}_1]}_{\underset{X \perp Z^+}{=} 0} E[\bar{X}]}} \quad + E[\hat{\beta}_1 x] \underbrace{E[\hat{\beta}_2 \hat{\alpha}_1 \bar{X}]}_{\underset{\mathcal{S} \cap \mathcal{D} = \emptyset}{=} E[\hat{\beta}_2] \underbrace{E[\hat{\alpha}_1 \bar{X}]}_{= \underbrace{E[\hat{\alpha}_1]}_{\underset{X \perp Z^+}{=} 0} E[\bar{X}]}}$$

$$+ \quad \underbrace{E[\hat{\beta}_2 \overline{Z^+} \hat{\beta}_2 \hat{\alpha}_1 x]}_{\underset{\mathcal{S} \cap \mathcal{D} = \emptyset}{=} x E[\hat{\beta}_2 \hat{\beta}_2] \underbrace{E[\overline{Z^+} \hat{\alpha}_1]}_{= E[\overline{Z^+}] \underbrace{E[\hat{\alpha}_1]}_{\underset{X \perp Z^+}{=} 0}}} \quad - E[\hat{\beta}_2 \overline{Z^+}] \underbrace{E[\hat{\beta}_2 \hat{\alpha}_1 x]}_{\underset{\mathcal{S} \cap \mathcal{D} = \emptyset}{=} x E[\hat{\beta}_2] \underbrace{E[\hat{\alpha}_1]}_{\underset{X \perp Z^+}{=} 0}}$$

$$- \quad \underbrace{E[\hat{\beta}_2 \overline{Z^+} \hat{\beta}_2 \hat{\alpha}_1 \bar{X}]}_{\underset{\mathcal{S} \cap \mathcal{D} = \emptyset}{=} E[\hat{\beta}_2 \hat{\beta}_2] \underbrace{E[\overline{Z^+} \hat{\alpha}_1 \bar{X}]}_{= 0}} \quad + E[\hat{\beta}_2 \overline{Z^+}] \underbrace{E[\hat{\beta}_2 \hat{\alpha}_1 \bar{X}]}_{\underset{\mathcal{S} \cap \mathcal{D} = \emptyset}{=} E[\hat{\beta}_2] \underbrace{E[\hat{\alpha}_1 \bar{X}]}_{= \underbrace{E[\hat{\alpha}_1]}_{\underset{X \perp Z^+}{=} 0} E[\bar{X}]}} \quad )$$

$$= Var[\hat{\beta}_2 \hat{\alpha}_1 (x - \bar{X})]$$

$$\geq 0 \ ,$$

where we used $Cov[\bar{X}, \hat{\alpha}_1] = 0$ and $Cov[\overline{Z^+}, \hat{\alpha}_1] = 0$ exploiting $E[\bar{X}\hat{\alpha}_1] = E[\bar{X}]E[\hat{\alpha}_1]$ and $E[\overline{Z^+}\hat{\alpha}_1] = E[\overline{Z^+}]E[\hat{\alpha}_1]$, as well as $E[\overline{Z^+}\hat{\alpha}_1\bar{X}] = 0$ which will be proven in the following.

We start with deriving $Cov[\bar{X}, \hat{\alpha}_1] = 0$. Denoting the unbiased empirical variance of $X$ by $\hat{\sigma}_X^2 = \frac{1}{|\mathcal{D}|-1} \sum_{j \in \mathcal{D}} (X_j - \bar{X})^2$, we have

$$
\begin{aligned}
Cov[\bar{X}, \hat{\alpha}_1] =& Cov\left[\bar{X}, \frac{1}{(|\mathcal{D}|-1)\hat{\sigma}_X^2} \sum_{j \in \mathcal{D}} (X_j - \bar{X})(Z_j^+ - \overline{Z^+})\right] \\
=& E\left[\bar{X} \frac{1}{(|\mathcal{D}|-1)\hat{\sigma}_X^2} \sum_{j \in \mathcal{D}} (X_j - \bar{X})(Z_j^+ - \overline{Z^+})\right] - E[\bar{X}] E\left[\frac{1}{(|\mathcal{D}|-1)\hat{\sigma}_X^2} \sum_{j \in \mathcal{D}} (X_j - \bar{X})(Z_j^+ - \overline{Z^+})\right] \\
=& E\left[\bar{X} \frac{1}{(|\mathcal{D}|-1)\hat{\sigma}_X^2} \sum_{j \in \mathcal{D}} (X_j - \bar{X}) \underbrace{E[Z_j^+ - \overline{Z^+} \mid X]}_{=E[Z_j^+ - \overline{Z^+}]=0}\right] \\
=& - E[\bar{X}] E\left[\frac{1}{(|\mathcal{D}|-1)\hat{\sigma}_X^2} \sum_{j \in \mathcal{D}} (X_j - \bar{X}) \underbrace{E[Z_j^+ - \overline{Z^+} \mid X]}_{=E[Z_j^+ - \overline{Z^+}]=0}\right] = 0
\end{aligned}
$$

due to the unbiasedness of the empirical mean.

Next, recall that by assumption $Z_i^+ = \mu_{z^+} + \xi_i$, where $\xi_i \overset{i.i.d.}{\sim} \mathcal{N}(0,1)$ implicating $\overline{Z^+} = \mu_{Z^+} + \frac{1}{|\mathcal{D}|} \sum_{i \in \mathcal{D}} \xi_i$. Hence, we can rewrite $Cov[\overline{Z^+}, \hat{\alpha}_1]$ as follows:

$$
\begin{aligned}
Cov[\overline{Z^+}, \hat{\alpha}_1] =& Cov\left[\overline{Z^+}, \frac{1}{(|\mathcal{D}|-1)\hat{\sigma}_X^2} \sum_{j \in \mathcal{D}} (X_j - \bar{X})(Z_j^+ - \overline{Z^+})\right] \\
=& Cov\left[\bar{\xi}, \frac{1}{(|\mathcal{D}|-1)\hat{\sigma}_X^2} \sum_{j \in \mathcal{D}} (X_j - \bar{X})(\xi_j - \bar{\xi})\right] \\
=& E\left[\bar{\xi} \frac{1}{(|\mathcal{D}|-1)\hat{\sigma}_X^2} \sum_{j \in \mathcal{D}} (X_j - \bar{X})(\xi_j - \bar{\xi})\right] - \underbrace{\underbrace{E[\bar{\xi}]}_{=0} E\left[\frac{1}{(|\mathcal{D}|-1)\hat{\sigma}_X^2} \sum_{j \in \mathcal{D}} (X_j - \bar{X})(\xi_j - \bar{\xi})\right]}_{=0} \\
=& \sum_{j \in \mathcal{D}} E\left[\left(\underbrace{\frac{X_j}{(|\mathcal{D}|-1)\hat{\sigma}_X^2}}_{=:\tilde{X}_j} - \frac{1}{|\mathcal{D}|} \sum_{i \in \mathcal{D}} \underbrace{\frac{X_i}{(|\mathcal{D}|-1)\hat{\sigma}_X^2}}_{=:\tilde{X}_i}\right)(\xi_j - \bar{\xi})\bar{\xi}\right] \\
=& \sum_{j \in \mathcal{D}} \underbrace{E\left[\left(\tilde{X}_j - \frac{1}{|\mathcal{D}|} \sum_{i \in \mathcal{D}} \tilde{X}_i\right)(\xi_j - \bar{\xi})\bar{\xi}\right]}_{X \perp\!\!\!\perp \xi \underbrace{E[\tilde{X}_j - \bar{\tilde{X}}]}_{=0} E[(\xi_j - \bar{\xi})\bar{\xi}]} = 0
\end{aligned}
$$

The last line follows due to centrality of $\xi$, $X \perp\!\!\!\perp \xi$ and unbiasedness of the empirical mean. Last, it remains to show that $E[\bar{X}\hat{\alpha}_1\overline{Z^+}]$ is zero.

$$
\begin{aligned}
E[\bar{X}\hat{\alpha}_1\overline{Z^+}] =& E\left[\bar{X}\frac{1}{(\mid \mathcal{D}\mid -1)\hat{\sigma}_X^2}\sum_{j\in\mathcal{D}}(X_j-\bar{X})(Z_j^+-\overline{Z^+})\overline{Z^+}\right] \\
=& \sum_{j\in\mathcal{D}}E\left[\frac{\bar{X}}{(\mid \mathcal{D}\mid -1)\hat{\sigma}_X^2}(X_j-\bar{X})(Z_j^+-\overline{Z^+})\overline{Z^+}\right] \\
=& \sum_{j\in\mathcal{D}}\underbrace{E\left[\frac{\bar{X}}{(\mid \mathcal{D}\mid -1)\hat{\sigma}_X^2}(X_j-\bar{X})\right]}_{=E\left[\underbrace{\frac{X_j\bar{X}}{(\mid \mathcal{D}\mid -1)\hat{\sigma}_X^2}}_{=:\check{X}_j}-\frac{1}{|\mathcal{D}|}\sum_{i\in\mathcal{D}}\underbrace{\frac{X_i\bar{X}}{(\mid \mathcal{D}\mid -1)\hat{\sigma}_X^2}}_{=:\check{X}_i}\right]}\ \ E[(Z_j^+-\overline{Z^+})\overline{Z^+}] \\
=& \sum_{j\in\mathcal{D}}\underbrace{E[\check{X}_j-\bar{\check{X}}]}_{=0}E[(Z_j^+-\overline{Z^+})\overline{Z^+}]=0\ ,
\end{aligned}
$$

Here, we used $X \perp\!\!\!\perp Z^+$ and the unbiasedness of the empirical mean, which concludes the proof.

$\square$

# B  Additional Experiments

In the following section, we show additional experiments to support the results discussed in Section 4.

Simulations were run in *R (version 4.4.3)*.

R Core Team (2025). R: A Language and Environment for Statistical Computing. R Foundation for Statistical Computing, Vienna, Austria. https://www.R-project.org/.

– *ggplot2* was used for data visualization.

Wickham, H. (2016), ggplot2: Elegant Graphics for Data Analysis. Springer-Verlag New York.

– *glmnet* was used to perform ridge regression.

Friedman J., Tibshirani R., Hastie T. (2010). Regularization Paths for Generalized Linear Models via Coordinate Descent. Journal of Statistical Software, 33(1), 1-22.

The simulations were run locally on a laptop (12th Gen Intel Core i7, 32 GB) and the time required to run all simulations was around 2.5 hours.

## B.1  Variance

Below, in Figures 6–9 we show the results for the quadratic and the linear model for RR and TSR as well as its versions with ridge penalty for the first of the two regression steps, respectively. For the boxplots, we differentiated between two cases. One is based on the biased dataset $\mathcal{S}$ and the other on the unbiased dataset $\mathcal{D}$. On the right hand side, we show the 95%-areas and mean of the estimations. For completeness, we accompany these results by providing the numerical values for them in Table 1 and Table 2.

The mean and standard deviation of the MSE remain smaller in $\mathcal{S}$ than $\mathcal{D}$, as one would expect, since the first regression was performed based only on the data underlying selection. For TSR and RR, the mean and standard deviation of MSE decrease for increasing $n$. For the naive estimator, this effect is not that pronounced. Mean and standard deviation of MSE are smaller for TSR than for RR, whereby the difference also vanishes when $n$ increases. Furthermore, we observe that mean and standard deviation of the MSE of the RR and TSR estimator do not differ distinctly between OLS and ridge regression, respectively. Last, there were no clear differences between $\mathcal{S}\cap\mathcal{D}=\emptyset$ and $\mathcal{S}\subset\mathcal{D}$ recognizable.

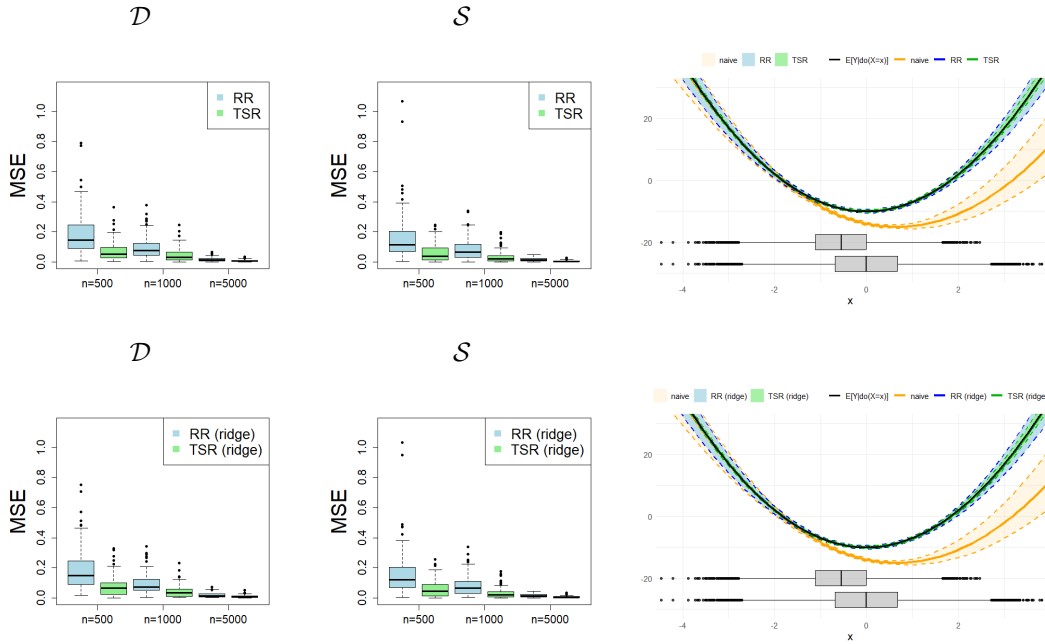

Figure 6: **Quadratic model**: Boxplots of the MSE over $\mathcal{D}$ and $\mathcal{S}$ ($\mathcal{S} \subset \mathcal{D}$) of RR and TSR for $n \in \{500, 1000, 5000\}$. The plots on the right show the associated $95\%$-areas of naive, RR and TSR estimation for $n = 500$. The upper boxplot represents the distribution of X in $\mathcal{S}$ and the lower in $\mathcal{D}$. The curves for naive, RR and TSR display the mean estimation over the simulation runs.

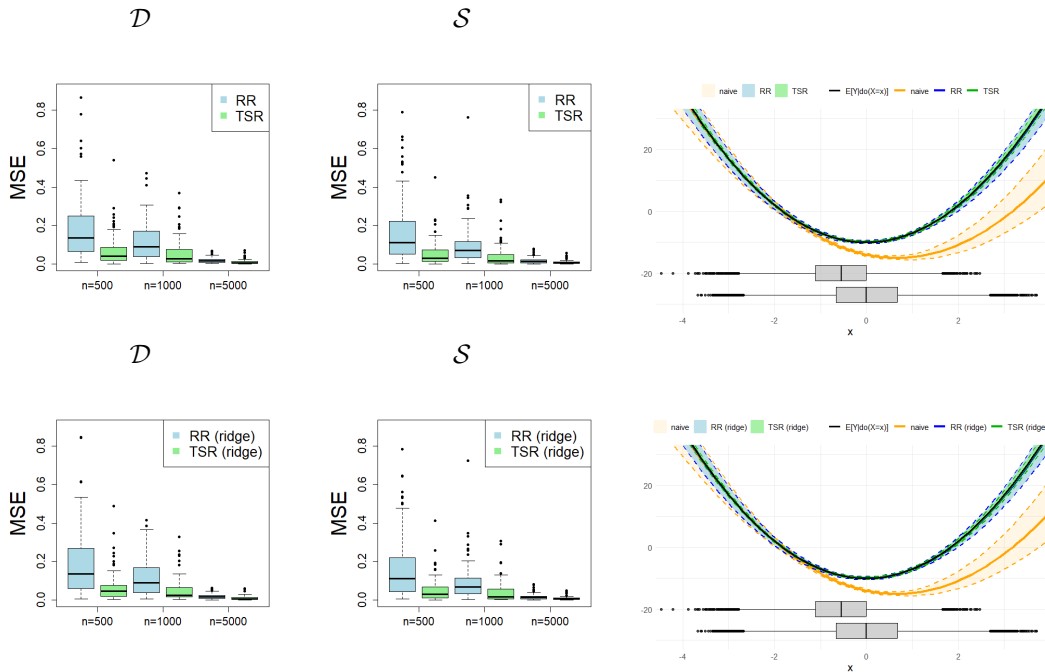

Figure 7: **Quadratic model**: Boxplots of the MSE over $\mathcal{D}$ and $\mathcal{S}$ ($\mathcal{S} \cap \mathcal{D} = \emptyset$) of RR and TSR for $n \in \{500, 1000, 5000\}$. The plots on the right show the associated $95\%$-areas of naive, RR and TSR estimation for $n = 500$. The upper boxplot represents the distribution of X in $\mathcal{S}$ and the lower in $\mathcal{D}$. The curves for naive, RR and TSR display the mean estimation over the simulation runs.

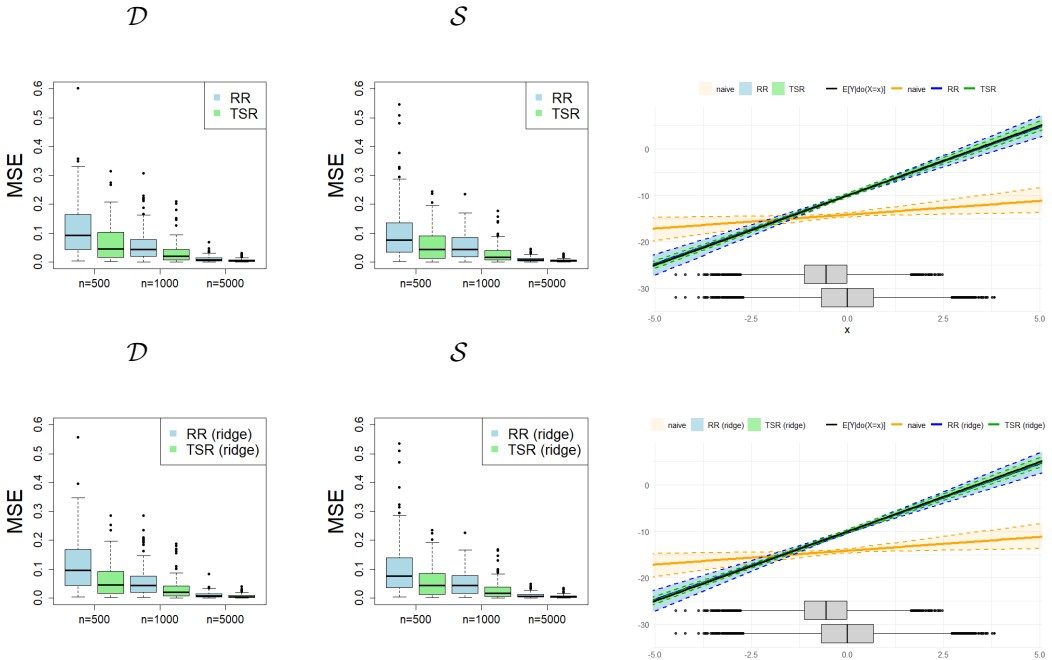

Figure 8: **Linear model**: Boxplots of the MSE over $\mathcal{D}$ and $\mathcal{S}$ ($\mathcal{S} \subset \mathcal{D}$) of RR and TSR for $n \in \{500, 1000, 5000\}$. The plots on the right show the associated $95\%$-areas of naive, RR and TSR estimation for $n = 500$. The upper boxplot represents the distribution of X in $\mathcal{S}$ and the lower in $\mathcal{D}$. The curves for naive, RR and TSR display the mean estimation over the simulation runs.

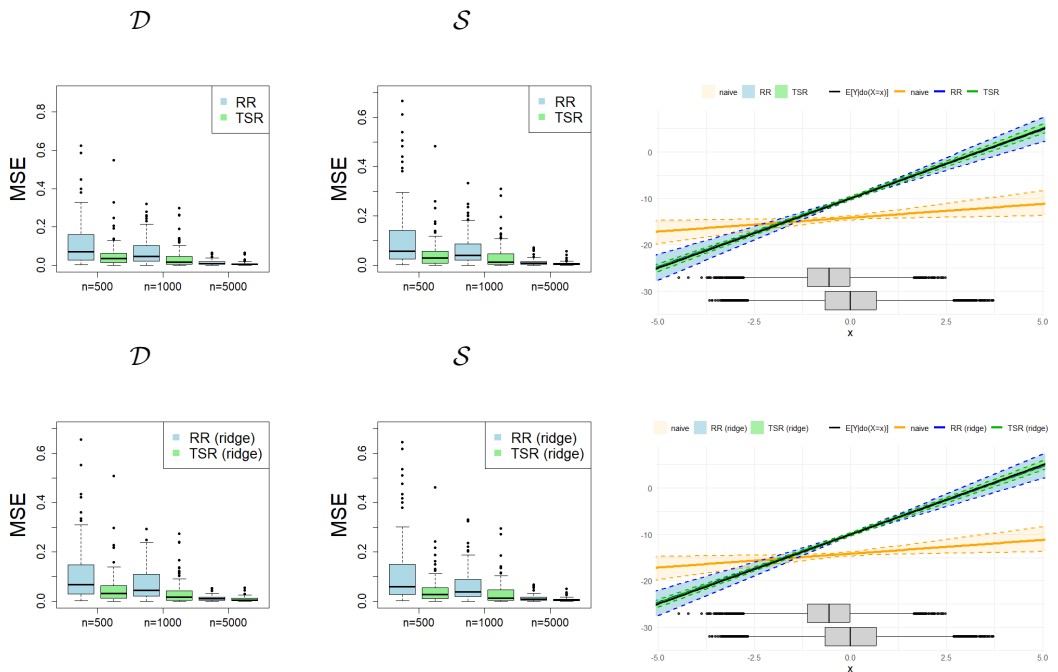

Figure 9: **Linear model**: Boxplots of the MSE over $\mathcal{D}$ and $\mathcal{S}$ ($\mathcal{S} \cap \mathcal{D} = \emptyset$) of RR and TSR for $n \in \{500, 1000, 5000\}$. The plots on the right show the associated $95\%$-areas of naive, RR and TSR estimation for $n = 500$. The upper boxplot represents the distribution of X in **S** and the lower in $\mathcal{D}$. The curves for naive, RR and TSR display the mean estimation over the simulation runs.

Table 1: **Quadratic model**: mean (sd) of MSE over $\mathcal{S}$ and $\mathcal{D}$.

| | $\mathcal{S}$ | | | | | |
|---|---|---|---|---|---|---|
| | $\mathcal{S} \subset \mathcal{D}$ | | | $\mathcal{S} \cap \mathcal{D} = \emptyset$ | | |
| | $n = 500$ | $n = 1000$ | $n = 5000$ | $n = 500$ | $n = 1000$ | $n = 5000$ |
| naive | 12.35 (1.75) | 12.00 (1.22) | 11.96 (0.48) | 12.53 (1.72) | 11.94 (1.15) | 11.90 (0.47) |
| RR | 0.16 (0.16) | 0.08 (0.07) | 0.02 (0.01) | 0.17 (0.17) | 0.09 (0.10) | 0.02 (0.02) |
| RR (ridge) | 0.16 (0.16) | 0.08 (0.07) | 0.02 (0.01) | 0.17 (0.17) | 0.09 (0.10) | 0.02 (0.02) |
| TSR | 0.06 (0.06) | 0.03 (0.04) | 0.01 (0.01) | 0.05 (0.06) | 0.04 (0.06) | 0.01 (0.01) |
| TSR (ridge) | 0.06 (0.06) | 0.03 (0.04) | 0.01 (0.01) | 0.05 (0.06) | 0.04 (0.05) | 0.01 (0.01) |

| | $\mathcal{D}$ | | | | | |
|---|---|---|---|---|---|---|
| | $\mathcal{S} \subset \mathcal{D}$ | | | $\mathcal{S} \cap \mathcal{D} = \emptyset$ | | |
| | $n = 500$ | $n = 1000$ | $n = 5000$ | $n = 500$ | $n = 1000$ | $n = 5000$ |
| naive | 32.78 (4.68) | 31.16 (3.68) | 30.93 (1.46) | 32.33 (4.79) | 31.07 (3.40) | 30.93 (1.66) |
| RR | 0.19 (0.14) | 0.10 (0.08) | 0.02 (0.01) | 0.19 (0.17) | 0.12 (0.10) | 0.02 (0.01) |
| RR (ridge) | 0.19 (0.14) | 0.09 (0.07) | 0.02 (0.01) | 0.19 (0.17) | 0.11 (0.09) | 0.02 (0.01) |
| TSR | 0.07 (0.07) | 0.05 (0.05) | 0.01 (0.01) | 0.07 (0.08) | 0.05 (0.07) | 0.01 (0.01) |
| TSR (ridge) | 0.08 (0.07) | 0.04 (0.04) | 0.01 (0.01) | 0.07 (0.08) | 0.05 (0.06) | 0.01 (0.01) |

Table 2: **Linear model**: mean (sd) of MSE over $\mathcal{S}$ and $\mathcal{D}$.

| | $\mathcal{S}$ | | | | | |
|---|---|---|---|---|---|---|
| | $\mathcal{S} \subset \mathcal{D}$ | | | $\mathcal{S} \cap \mathcal{D} = \emptyset$ | | |
| | $n = 500$ | $n = 1000$ | $n = 5000$ | $n = 500$ | $n = 1000$ | $n = 5000$ |
| naive | 11.91 (1.57) | 11.52 (1.13) | 11.59 (0.46) | 12.07 (1.54) | 11.48 (1.10) | 11.56 (0.46) |
| RR | 0.11 (0.11) | 0.05 (0.05) | 0.01 (0.01) | 0.12 (0.14) | 0.06 (0.07) | 0.01 (0.01) |
| RR (ridge) | 0.11 (0.11) | 0.05 (0.05) | 0.01 (0.01) | 0.11 (0.14) | 0.06 (0.07) | 0.01 (0.01) |
| TSR | 0.06 (0.06) | 0.03 (0.04) | 0.01 (0.01) | 0.05 (0.07) | 0.04 (0.05) | 0.01 (0.01) |
| TSR (ridge) | 0.06 (0.06) | 0.03 (0.04) | 0.01 (0.01) | 0.05 (0.06) | 0.03 (0.05) | 0.01 (0.01) |

| | $\mathcal{D}$ | | | | | |
|---|---|---|---|---|---|---|
| | $\mathcal{S} \subset \mathcal{D}$ | | | $\mathcal{S} \cap \mathcal{D} = \emptyset$ | | |
| | $n = 500$ | $n = 1000$ | $n = 5000$ | $n = 500$ | $n = 1000$ | $n = 5000$ |
| naive | 23.52 (3.10) | 22.01 (2.21) | 22.38 (0.98) | 23.35 (3.08) | 22.02 (2.21) | 22.37 (1.05) |
| RR | 0.12 (0.10) | 0.06 (0.06) | 0.01 (0.01) | 0.11 (0.12) | 0.07 (0.07) | 0.01 (0.01) |
| RR (ridge) | 0.12 (0.10) | 0.06 (0.06) | 0.01 (0.01) | 0.11 (0.12) | 0.07 (0.07) | 0.01 (0.01) |
| TSR | 0.06 (0.06) | 0.03 (0.04) | 0.01 (0.01) | 0.05 (0.07) | 0.04 (0.06) | 0.01 (0.01) |
| TSR (ridge) | 0.07 (0.06) | 0.03 (0.04) | 0.01 (0.01) | 0.05 (0.07) | 0.04 (0.05) | 0.01 (0.01) |

## B.2 Examples with selection bias as well as confounding

In this section, we will present further details concerning the six examples mentioned in Section 4.1. First, we state the data generating processes. Then, for each example, we show six plots (Figures 10–15), which show the 95%-areas and means of TSR and RR for varying $n \in \{500, 1000, 5000\}$ and the effect of adding a ridge penalty. Finally, we will present the mean and standard deviation of the MSE over all considered settings, evaluated on $\mathcal{S}$ as well as on $\mathcal{D}$ in Tables 3–8.

The data generating processes are given in the following.

 Example 1:

$$\varepsilon_X, \varepsilon_Y \sim \mathcal{N}(0,1)$$
$$Z \sim \mathcal{N}(-2,1)$$
$$X := 2Z + \varepsilon_X$$
$$S := \mathbf{1}_{\{X+Z<-6\}}$$
$$Y := 0.2X^2 + 5Z + \varepsilon_Y$$

$$E[Y \mid X = x] = 0.2x^2 - 2 + 2x$$
$$E[Y \mid do(X = x)] = 0.2x^2 - 10$$

 Example 2:

$$\varepsilon_X, \varepsilon_Y \sim \mathcal{N}(0,1)$$
$$Z \sim \mathcal{N}(-1,4)$$
$$X := Z + \varepsilon_X$$
$$S \sim \mathcal{B}ern\left(\frac{1}{(1 + exp(-X))(1 + exp(Z))}, n\right)$$
$$Y := X + 5Z + \varepsilon_Y$$

$$E[Y \mid X = x] = 5x - 1$$
$$E[Y \mid do(X = x)] = x - 5$$

 Example 3:

$$\varepsilon_X, \varepsilon_Y \sim \mathcal{N}(0,1)$$
$$W \sim \mathcal{N}(2, 0.3^2)$$
$$X := W + \varepsilon_X$$
$$Z \sim \mathcal{N}(-0.3, 1)$$
$$S := \mathbf{1}_{\{Z>0, X<9\}}$$
$$Y := 0.2X^2 + Z + 3W + \varepsilon_Y$$

$$E[Y \mid X = x] = 0.2x^2 + 5.7 + 3\left(\frac{0.3^2}{0.3^2 + 1}(x - 2)\right)$$
$$E[Y \mid do(X = x)] = 0.2x^2 - 0.3 + 6$$

 Example 4:

$$\varepsilon_X, \varepsilon_Y \sim \mathcal{N}(0,1)$$
$$W \sim \mathcal{N}(2, 0.3^2)$$
$$X := W + \varepsilon_X$$
$$Z \sim \mathcal{N}(0,1)$$
$$S \sim \mathcal{B}ern\left(\frac{1}{(1 + exp(X))(1 + exp(Z))}\right)$$
$$Y := 0.5X + Z + 3W + \varepsilon_Y$$

$$E[Y \mid X = x] = 0.5x + 6 + 3(0.3^4 + 0.3^2)(x - 2)$$
$$E[Y \mid do(X = x)] = 0.5x + 6$$

755 Example 5:

$$\varepsilon_X, \varepsilon_Z, \varepsilon_Y \sim \mathcal{N}(0,1)$$
$$W \sim \mathcal{N}(-1,1)$$
$$X := W + \varepsilon_X$$
$$Z := -2X + \varepsilon_Z$$
$$S \sim \mathcal{B}ern\left(\frac{1}{(1+exp(X))(1+exp(Z))}\right)$$
$$Y := X^2 + Z + 2W + \varepsilon_Y$$

756

$$E[Y \mid X = x] = x^2 - x - 1$$
$$E[Y \mid do(X = x)] = x^2 - 2x - 2$$

757 Example 6:

$$\varepsilon_X, \varepsilon_Z, \varepsilon_Y \sim \mathcal{N}(0,1)$$
$$W \sim \mathcal{N}(2,1)$$
$$X := W + \varepsilon_X$$
$$Z := X + \varepsilon_Z$$
$$S := \mathbf{1}_{\{(ZX)<1,(ZX)^2+Z>1\}}$$
$$Y := \frac{1}{10}(X + 5Z + 3W + \varepsilon_Y)$$

758

$$E[Y \mid X = x] = \frac{3}{10} + \frac{3}{4}x$$
$$E[Y \mid do(X = x)] = \frac{3}{5}(x + 1)$$

759 The results visualized in Figures 10–15, and Tables 3–8 show that the spread of RR and TSR gets
760 reduced by adding ridge penalty. But we also recognize that adding a ridge penalty can go along with
761 adding bias, which is evident for $n = 500$. Especially, for Example 1, the ridge estimation deviates
762 far from the true underlying causal effect, which is even outside $95\%$-area of TSR with ridge penalty.
763 The same applies to RR with a ridge regression penalty for Example 1, which in turn does not include
764 $E[Y \mid X]$, in its confidence interval, even though $E[Y \mid X]$ is the quantity RR aims to estimate. As
765 expected, RR does not recover $E[Y \mid do(X)]$ (since it is misspecified in these settings), which is
766 evident since the underlying causal effect is not covered by the $95\%$ of the RR estimator at least
767 in a wide range of the distribution of $X$ in $\mathcal{D}$. This is exactly what we would expect to see when
768 $E[Y \mid do(X)]$ and $E[Y \mid X]$ differ significantly. Of course, the estimates vary stronger the further
769 the particular values of $X$ are from the support of $X$ in $\mathcal{S}$. But again, the variation diminishes with
770 increasing sample size. Just to notice, particularly for Example 6, it becomes visible how unreliable
771 the naive estimation is. It suggests a quadratic relationship instead of the linear ground truth.

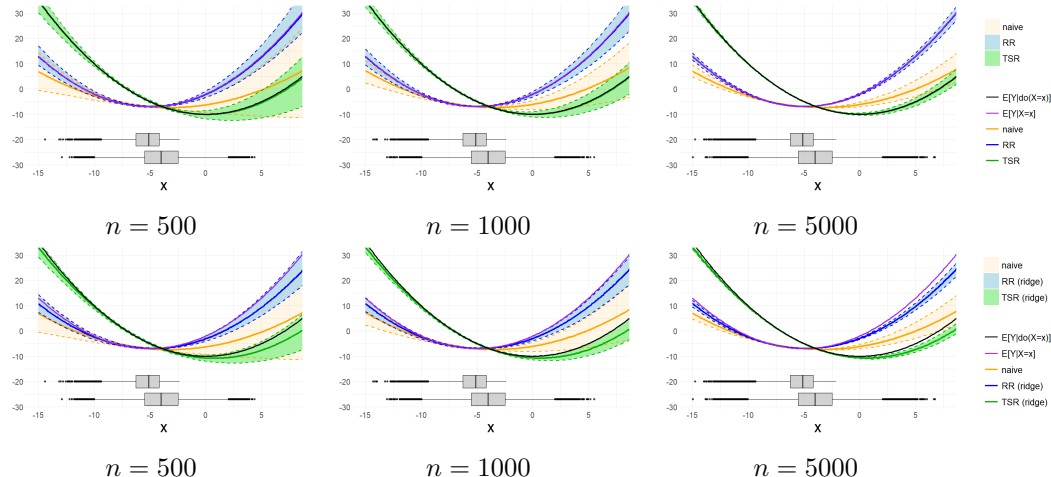

Figure 10: **Example 1**: Comparison of the central 95%-areas of naive, RR and TSR of the simulation runs for the DAG in Figure 4 (a) with sample size $n \in \{500, 1000, 5000\}$. The upper boxplot represents the distribution of $X$ in $\mathcal{S}$ and the lower in $\mathcal{D}$ ($\mathcal{S} \cap \mathcal{D} = \emptyset$). The curves for naive, RR and TSR display the mean estimation over the simulation runs.

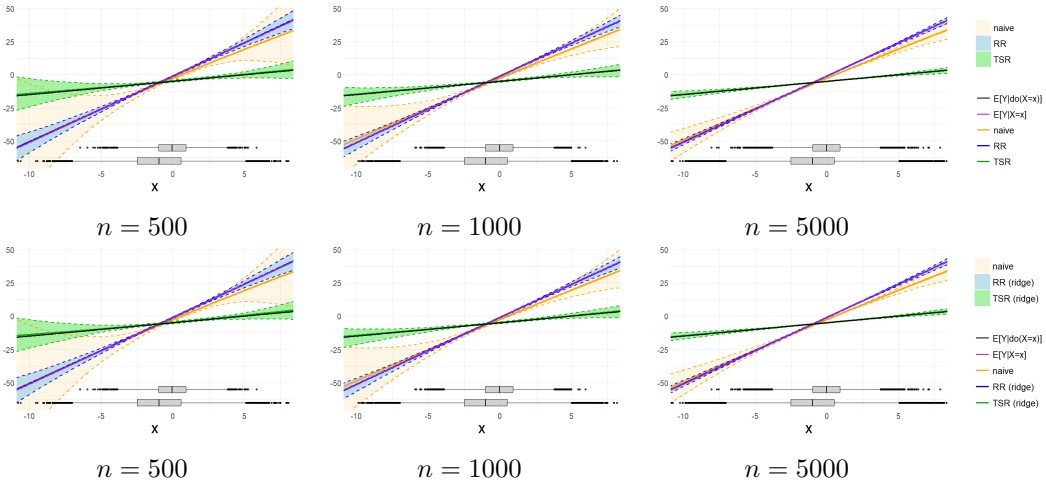

Figure 11: **Example 2**: Comparison of the central 95%-areas of naive, RR and TSR of the simulation runs for the DAG in Figure 4 (a) with sample size $n \in \{500, 1000, 5000\}$. The upper boxplot represents the distribution of $X$ in $\mathcal{S}$ and the lower in $\mathcal{D}$ ($\mathcal{S} \cap \mathcal{D} = \emptyset$). The curves for naive, RR and TSR display the mean estimation over the simulation runs.

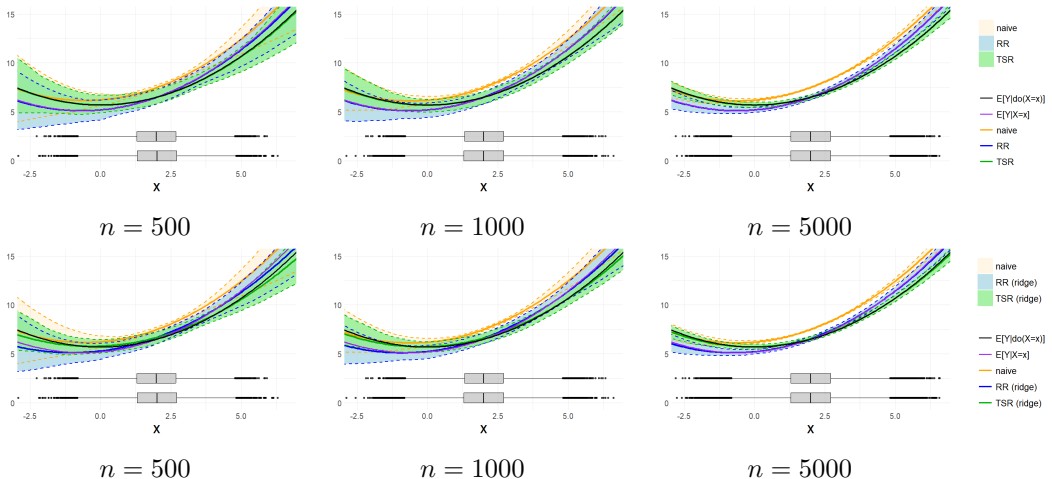

Figure 12: **Example 3**: Comparison of the central 95%-areas of naive, RR and TSR of the simulation runs for the DAG in Figure 4 (b) with sample size $n \in \{500, 1000, 5000\}$. The upper boxplot represents the distribution of $X$ in $\mathcal{S}$ and the lower in $\mathcal{D}$ ($\mathcal{S} \cap \mathcal{D} = \emptyset$). The curves for naive, RR and TSR display the mean estimation over the simulation runs.

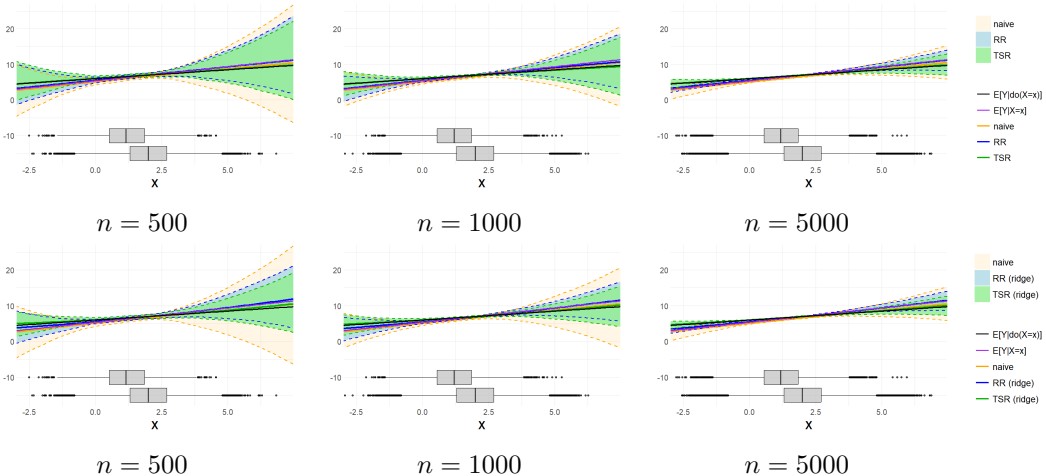

Figure 13: **Example 4**: Comparison of the central 95%-areas of naive, RR and TSR of the simulation runs for the DAG in Figure 4 (b) with sample size $n \in \{500, 1000, 5000\}$. The upper boxplot represents the distribution of $X$ in $\mathcal{S}$ and the lower in $\mathcal{D}$ ($\mathcal{S} \cap \mathcal{D} = \emptyset$). The curves for naive, RR and TSR display the mean estimation over the simulation runs.

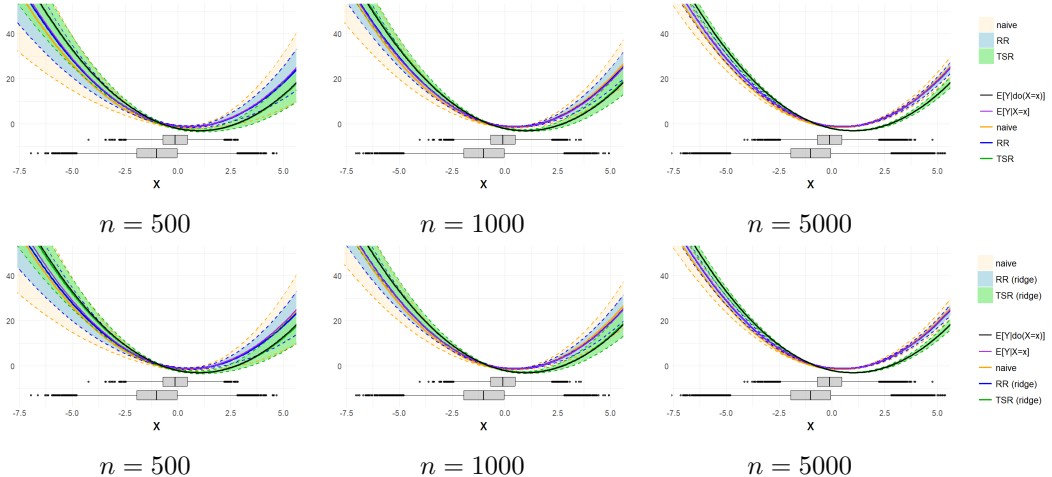

Figure 14: **Example 5**: Comparison of the central 95%-areas naive, RR and TSR of the simulation runs for the DAG in Figure 4 (c) with sample size $n \in \{500, 1000, 5000\}$. The upper boxplot represents the distribution of $X$ in $\mathcal{S}$ and the lower in $\mathcal{D}$ ($\mathcal{S} \cap \mathcal{D} = \emptyset$). The curves for naive, RR and TSR display the mean estimation over the simulation runs.

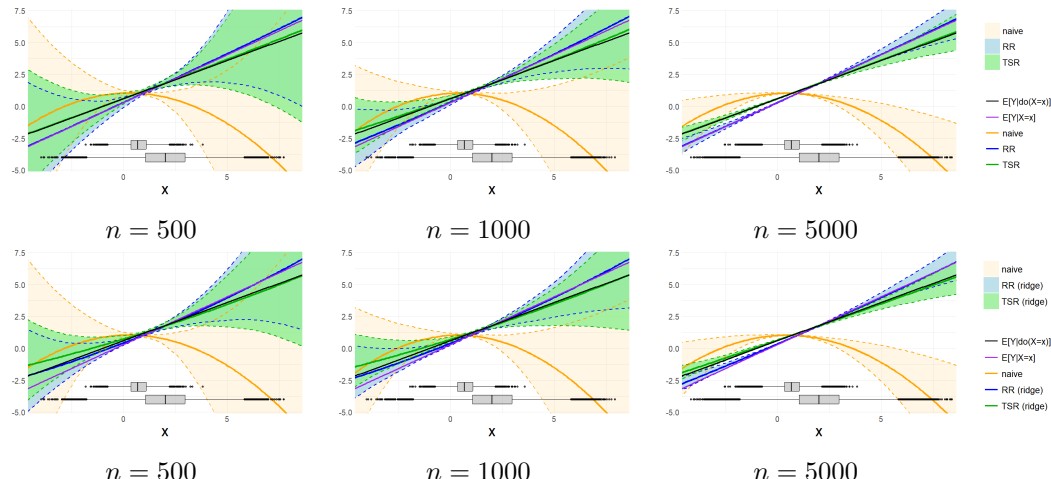

Figure 15: **Example 6**: Comparison of the central 95%-areas of naive, RR and TSR of the simulation runs for the DAG in Figure 4 (c) with sample size $n \in \{500, 1000, 5000\}$. The upper boxplot represents the distribution of $X$ in $\mathcal{S}$ and the lower in $\mathcal{D}$ ($\mathcal{S} \cap \mathcal{D} = \emptyset$). The curves for naive, RR and TSR display the mean estimation over the simulation runs.

Table 3: **Example 1**: mean(sd) of MSE over $\mathcal{S}$ and $\mathcal{D}$.

| | $\mathcal{S}$ | | | | | |
|---|---|---|---|---|---|---|
| | $\mathcal{S} \subset \mathcal{D}$ | | | $\mathcal{S} \cap \mathcal{D} = \emptyset$ | | |
| | $n = 500$ | $n = 1000$ | $n = 5000$ | $n = 500$ | $n = 1000$ | $n = 5000$ |
| naive | 15.89 (1.94) | 15.92 (1.55) | 15.91 (0.61) | 15.94 (1.95) | 15.91 (1.37) | 15.90 (0.58) |
| RR | 16.20 (1.81) | 16.25 (1.46) | 16.26 (0.58) | 16.26 (1.82) | 16.19 (1.37) | 16.28 (0.57) |
| RR (ridge) | 16.20 (1.86) | 16.23 (1.51) | 16.25 (0.59) | 16.27 (1.88) | 16.18 (1.40) | 16.27 (0.59) |
| TSR | 0.08 (0.10) | 0.05 (0.07) | 0.01 (0.01) | 0.07 (0.08) | 0.04 (0.05) | 0.01 (0.01) |
| TSR (ridge) | 0.08 (0.09) | 0.05 (0.07) | 0.01 (0.04) | 0.07 (0.07) | 0.04 (0.04) | 0.02 (0.01) |

| | $\mathcal{D}$ | | | | | |
|---|---|---|---|---|---|---|
| | $\mathcal{S} \subset \mathcal{D}$ | | | $\mathcal{S} \cap \mathcal{D} = \emptyset$ | | |
| | $n = 500$ | $n = 1000$ | $n = 5000$ | $n = 500$ | $n = 1000$ | $n = 5000$ |
| naive | 12.37 (2.33) | 12.37 (1.50) | 12.17 (0.69) | 12.36 (2.27) | 12.37 (1.49) | 12.16 (0.66) |
| RR | 19.90 (2.27) | 19.94 (1.58) | 20.00 (0.83) | 19.62 (2.19) | 19.95 (1.85) | 20.00 (0.70) |
| RR (ridge) | 17.78 (1.87) | 17.89 (1.34) | 17.95 (0.69) | 17.59 (1.85) | 17.88 (1.53) | 17.94 (0.59) |
| TSR | 0.16 (0.15) | 0.08 (0.09) | 0.01 (0.01) | 0.16 (0.16) | 0.08 (0.07) | 0.01 (0.01) |
| TSR (ridge) | 0.21 (0.18) | 0.14 (0.13) | 0.08 (0.04) | 0.20 (0.20) | 0.13 (0.10) | 0.09 (0.05) |

Table 4: **Example 2**: mean(sd) of MSE over $\mathcal{S}$ and $\mathcal{D}$.

| | $\mathcal{S}$ | | | | | |
|---|---|---|---|---|---|---|
| | $\mathcal{S} \subset \mathcal{D}$ | | | $\mathcal{S} \cap \mathcal{D} = \emptyset$ | | |
| | $n = 500$ | $n = 1000$ | $n = 5000$ | $n = 500$ | $n = 1000$ | $n = 5000$ |
| naive | 31.64 (7.52) | 30.77 (4.99) | 30.92 (2.29) | 31.89 (8.10) | 31.13 (5.51) | 30.68 (2.17) |
| RR | 47.27 (8.47) | 47.72 (5.23) | 48.17 (2.37) | 48.86 (9.22) | 48.10 (4.88) | 48.16 (2.90) |
| RR (ridge) | 46.90 (8.33) | 47.57 (5.21) | 48.06 (2.37) | 48.51 (9.13) | 47.94 (4.87) | 48.05 (2.89) |
| TSR | 0.30 (0.31) | 0.15 (0.22) | 0.03 (0.04) | 0.22 (0.25) | 0.14 (0.15) | 0.02 (0.02) |
| TSR (ridge) | 0.30 (0.30) | 0.15 (0.22) | 0.03 (0.04) | 0.23 (0.25) | 0.14 (0.15) | 0.02 (0.02) |

| | $\mathcal{D}$ | | | | | |
|---|---|---|---|---|---|---|
| | $\mathcal{S} \subset \mathcal{D}$ | | | $\mathcal{S} \cap \mathcal{D} = \emptyset$ | | |
| | $n = 500$ | $n = 1000$ | $n = 5000$ | $n = 500$ | $n = 1000$ | $n = 5000$ |
| naive | 68.41 (17.95) | 65.60 (12.98) | 64.47 (5.40) | 68.59 (23.01) | 64.20 (12.74) | 63.82 (6.02) |
| RR | 80.62 (8.17) | 79.97 (5.59) | 80.11 (2.41) | 79.39 (7.54) | 80.18 (5.53) | 79.88 (2.50) |
| RR (ridge) | 80.28 (8.15) | 79.79 (5.58) | 79.99 (2.42) | 79.01 (7.53) | 80.00 (5.51) | 79.76 (2.50) |
| TSR | 0.53 (0.59) | 0.24 (0.26) | 0.04 (0.05) | 0.46 (0.48) | 0.22 (0.21) | 0.04 (0.04) |
| TSR (ridge) | 0.53 (0.58) | 0.24 (0.26) | 0.04 (0.05) | 0.46 (0.48) | 0.22 (0.21) | 0.04 (0.04) |

Table 5: **Example 3**: mean(sd) of MSE over $\mathcal{S}$ and $\mathcal{D}$.

|  | $\mathcal{S}$ | | | | | |
| --- | --- | --- | --- | --- | --- | --- |
|  | $\mathcal{S} \subset \mathcal{D}$ | | | $\mathcal{S} \cap \mathcal{D} = \emptyset$ | | |
|  | $n = 500$ | $n = 1000$ | $n = 5000$ | $n = 500$ | $n = 1000$ | $n = 5000$ |
| naive | 1.09 (0.21) | 1.08 (0.15) | 1.07 (0.07) | 1.08 (0.22) | 1.08 (0.15) | 1.07 (0.07) |
| RR | 0.26 (0.31) | 0.17 (0.13) | 0.09 (0.03) | 0.25 (0.31) | 0.18 (0.15) | 0.09 (0.03) |
| RR (ridge) | 0.22 (0.23) | 0.16 (0.11) | 0.09 (0.03) | 0.21 (0.23) | 0.17 (0.13) | 0.09 (0.03) |
| TSR | 0.19 (0.31) | 0.10 (0.13) | 0.02 (0.02) | 0.18 (0.31) | 0.10 (0.15) | 0.02 (0.02) |
| TSR (ridge) | 0.16 (0.23) | 0.09 (0.11) | 0.02 (0.02) | 0.15 (0.24) | 0.10 (0.13) | 0.02 (0.02) |

|  | $\mathcal{D}$ | | | | | |
| --- | --- | --- | --- | --- | --- | --- |
|  | $\mathcal{S} \subset \mathcal{D}$ | | | $\mathcal{S} \cap \mathcal{D} = \emptyset$ | | |
|  | $n = 500$ | $n = 1000$ | $n = 5000$ | $n = 500$ | $n = 1000$ | $n = 5000$ |
| naive | 1.08 (0.21) | 1.08 (0.15) | 1.07 (0.07) | 1.08 (0.21) | 1.08 (0.14) | 1.07 (0.07) |
| RR | 0.26 (0.31) | 0.17 (0.13) | 0.09 (0.03) | 0.25 (0.31) | 0.18 (0.15) | 0.09 (0.03) |
| RR (ridge) | 0.22 (0.23) | 0.16 (0.11) | 0.09 (0.03) | 0.21 (0.23) | 0.17 (0.13) | 0.09 (0.03) |
| TSR | 0.19 (0.31) | 0.10 (0.13) | 0.02 (0.02) | 0.18 (0.31) | 0.10 (0.15) | 0.02 (0.02) |
| TSR (ridge) | 0.16 (0.23) | 0.09 (0.11) | 0.02 (0.02) | 0.15 (0.24) | 0.10 (0.13) | 0.02 (0.02) |

Table 6: **Example 4**: mean(sd) of MSE over $\mathcal{S}$ and $\mathcal{D}$.

|  | $\mathcal{S}$ | | | | | |
| --- | --- | --- | --- | --- | --- | --- |
|  | $\mathcal{S} \subset \mathcal{D}$ | | | $\mathcal{S} \cap \mathcal{D} = \emptyset$ | | |
|  | $n = 500$ | $n = 1000$ | $n = 5000$ | $n = 500$ | $n = 1000$ | $n = 5000$ |
| naive | 0.62 (0.44) | 0.51 (0.26) | 0.45 (0.11) | 0.62 (0.42) | 0.51 (0.25) | 0.45 (0.11) |
| RR | 0.23 (0.19) | 0.17 (0.13) | 0.10 (0.04) | 0.23 (0.22) | 0.16 (0.12) | 0.11 (0.03) |
| RR (ridge) | 0.20 (0.16) | 0.16 (0.13) | 0.10 (0.04) | 0.20 (0.18) | 0.15 (0.12) | 0.11 (0.03) |
| TSR | 0.15 (0.17) | 0.06 (0.06) | 0.01 (0.01) | 0.15 (0.15) | 0.06 (0.06) | 0.01 (0.01) |
| TSR (ridge) | 0.11 (0.14) | 0.05 (0.05) | 0.01 (0.01) | 0.11 (0.12) | 0.05 (0.05) | 0.01 (0.01) |

|  | $\mathcal{D}$ | | | | | |
| --- | --- | --- | --- | --- | --- | --- |
|  | $\mathcal{S} \subset \mathcal{D}$ | | | $\mathcal{S} \cap \mathcal{D} = \emptyset$ | | |
|  | $n = 500$ | $n = 1000$ | $n = 5000$ | $n = 500$ | $n = 1000$ | $n = 5000$ |
| naive | 0.71 (0.74) | 0.41 (0.26) | 0.28 (0.08) | 0.73 (0.76) | 0.42 (0.26) | 0.28 (0.08) |
| RR | 0.31 (0.40) | 0.16 (0.15) | 0.09 (0.05) | 0.32 (0.38) | 0.16 (0.16) | 0.09 (0.05) |
| RR (ridge) | 0.25 (0.28) | 0.15 (0.12) | 0.09 (0.05) | 0.25 (0.26) | 0.15 (0.13) | 0.09 (0.05) |
| TSR | 0.24 (0.31) | 0.10 (0.13) | 0.02 (0.02) | 0.24 (0.30) | 0.10 (0.14) | 0.02 (0.02) |
| TSR (ridge) | 0.17 (0.20) | 0.08 (0.08) | 0.02 (0.02) | 0.17 (0.20) | 0.08 (0.09) | 0.02 (0.02) |

Table 7: **Example 5**: mean(sd) of MSE over $\mathcal{S}$ and $\mathcal{D}$.

| | $\mathcal{S}$ | | | | | |
| | $\mathcal{S} \subset \mathcal{D}$ | | | $\mathcal{S} \cap \mathcal{D} = \emptyset$ | | |
| | $n = 500$ | $n = 1000$ | $n = 5000$ | $n = 500$ | $n = 1000$ | $n = 5000$ |
|---|---|---|---|---|---|---|
| naive | 1.72 (0.76) | 1.69 (0.61) | 1.46 (0.21) | 1.70 (0.74) | 1.61 (0.54) | 1.47 (0.22) |
| RR | 1.76 (0.62) | 1.70 (0.36) | 1.60 (0.16) | 1.77 (0.59) | 1.63 (0.31) | 1.60 (0.16) |
| RR (ridge) | 1.73 (0.60) | 1.68 (0.35) | 1.59 (0.16) | 1.73 (0.56) | 1.62 (0.30) | 1.59 (0.16) |
| TSR | 0.12 (0.09) | 0.06 (0.06) | 0.01 (0.01) | 0.12 (0.11) | 0.05 0.05) | 0.01 (0.01) |
| TSR (ridge) | 0.12 (0.10) | 0.06 (0.06) | 0.01 (0.01) | 0.12 (0.11) | 0.05 (0.05) | 0.01 (0.01) |

| | $\mathcal{D}$ | | | | | |
| | $\mathcal{S} \subset \mathcal{D}$ | | | $\mathcal{S} \cap \mathcal{D} = \emptyset$ | | |
| | $n = 500$ | $n = 1000$ | $n = 5000$ | $n = 500$ | $n = 1000$ | $n = 5000$ |
|---|---|---|---|---|---|---|
| naive | 5.79 (4.36) | 4.35 (2.93) | 3.93 (1.12) | 5.82 (4.41) | 4.33 (2.94) | 3.93 (1.15) |
| RR | 2.83 (1.97) | 2.19 (0.82) | 2.06 (0.39) | 2.87 (2.01) | 2.15 (0.75) | 2.06 (0.40) |
| RR (ridge) | 3.10 (2.33) | 2.29 (0.86) | 2.09 (0.40) | 3.13 (2.28) | 2.24 (0.77) | 2.09 (0.41) |
| TSR | 0.63 (0.82) | 0.32 (0.59) | 0.04 (0.06) | 0.65 (0.82) | 0.29 (0.51) | 0.05 (0.06) |
| TSR (ridge) | 0.69 (1.05) | 0.31 (0.56) | 0.04 (0.06) | 0.69 (0.97) | 0.28 (0.48) | 0.05 (0.07) |

Table 8: **Example 6**: mean(sd) of MSE over $\mathcal{S}$ and $\mathcal{D}$.

| | $\mathcal{S}$ | | | | | |
| | $\mathcal{S} \subset \mathcal{D}$ | | | $\mathcal{S} \cap \mathcal{D} = \emptyset$ | | |
| | $n = 500$ | $n = 1000$ | $n = 5000$ | $n = 500$ | $n = 1000$ | $n = 5000$ |
|---|---|---|---|---|---|---|
| naive | 0.22 (0.13) | 0.21 (0.09) | 0.18 (0.03) | 0.21 (0.10) | 0.20 (0.09) | 0.18 (0.03) |
| RR | 0.07 (0.06) | 0.05 (0.02) | 0.05 (0.01) | 0.07 (0.06) | 0.05 (0.03) | 0.05 (0.01) |
| RR (ridge) | 0.03 (0.02) | 0.03 (0.02) | 0.02 (0.01) | 0.03 (0.02) | 0.03 (0.02) | 0.04 (0.01) |
| TSR | 0.02 (0.07) | 0.01 (0.01) | 0.00 (0.00) | 0.03 (0.08) | 0.01 (0.01) | 0.00 (0.00) |
| TSR (ridge) | 0.02 (0.03) | 0.01 (0.01) | 0.00 (0.00) | 0.02 (0.02) | 0.01 (0.01) | 0.00 (0.00) |

| | $\mathcal{D}$ | | | | | |
| | $\mathcal{S} \subset \mathcal{D}$ | | | $\mathcal{S} \cap \mathcal{D} = \emptyset$ | | |
| | $n = 500$ | $n = 1000$ | $n = 5000$ | $n = 500$ | $n = 1000$ | $n = 5000$ |
|---|---|---|---|---|---|---|
| naive | 4.52 (3.70) | 4.23 (2.60) | 3.35 (0.82) | 4.35 (3.57) | 4.28 (2.63) | 3.36 (0.82) |
| RR | 0.26 (0.67) | 0.10 (0.10) | 0.05 (0.03) | 0.26 (0.66) | 0.11 (0.10) | 0.06 (0.03) |
| RR (ridge) | 0.13 (0.22) | 0.09 (0.09) | 0.04 (0.02) | 0.13 (0.23) | 0.09 (0.10) | 0.04 (0.02) |
| TSR | 0.20 (0.59) | 0.06 (0.08) | 0.01 (0.01) | 0.20 (0.56) | 0.06 (0.08) | 0.01 (0.01) |
| TSR (ridge) | 0.12 (0.19) | 0.07 (0.11) | 0.01 (0.01) | 0.12 (0.18) | 0.07 (0.11) | 0.01 (0.01) |

## B.3 Motivating Example Continued

In comparison to the example with $n = 500$ in the main part of the paper (Section 4.3), Figure 16 shows the results for a larger sample size ($n = 2000$). We see that when increasing the sample size $n$, the estimations spread less. Further, adding a ridge penalty affects the estimation less than for a smaller sample size.

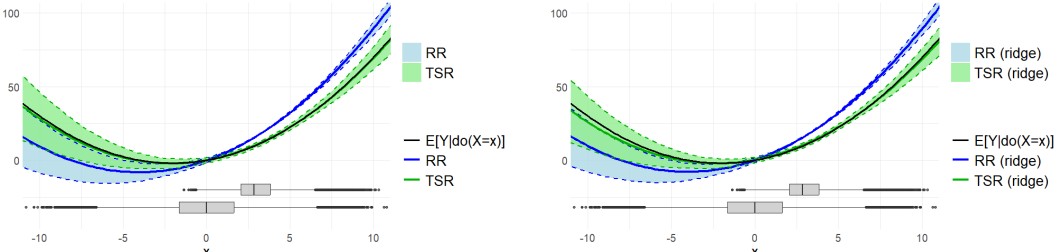

Figure 16: Comparison of the central 95%-areas of RR and TSR of the simulation runs for the DAG in Figure 2 (d) with sample size $n = 2000$. The upper boxplot represents the distribution of $X$ in $\mathcal{S}$ and the lower in $\mathcal{D}$ ($\mathcal{S} \cap \mathcal{D} = \emptyset$). The curves for RR and TSR display the mean estimation over the simulation runs.

## B.4 Robustness to Misspecification

In this section, we investigate the robustness of TSR to certain types of misspecification which is introduced by an unobserved latent variable $U$, as illustrated in Figure 17, where we stick to the scenario in which $\mathcal{S} \cap \mathcal{D} = \emptyset$. In case (a) $U$ is a cause of $Y$, in case (b) $U$ is a cause of $S$ and in case (c) $U$ is a confounder between $S$ and $Y$.

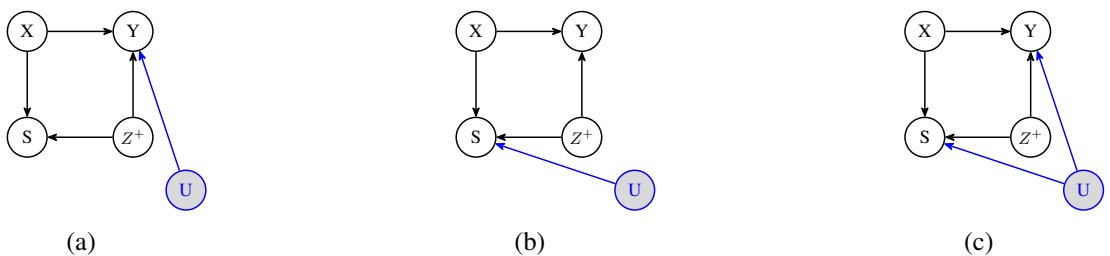

(a)  (b)  (c)

Figure 17: (a) unobserved cause of $S$, (b) unobserved regressor, (c) unobserved cause of $S$ and unobserved regressor

For case (a), our identifiability assumptions are still met (Theorem 2.4 still holds), but the missing regressor introduces a bias for our empirical estimator (i.e. Assumption 2.5 is violated). Hence, in the first regression, we estimate the $\beta$ coefficients biased, in the second step, we just calculate $\overline{Z^+}$ because we are not aware of $U$. However, as we do not estimate the $\beta$ for $U$, $\bar{U}$ cannot enter the estimation. For case (b), our assumptions are still fulfilled, however, our proxy $Z^+$ is weaker since $S$ has an additional unobserved cause. Last, for case (c), the PMAR assumption is violated. Hence, in addition to the bias entered through the missing regressor in (b), here the expression for $E[Y \mid do(X)]$ used to construct $\hat{\mu}_{TSR}(x)$ is wrong as its derivation that makes use of PMAR.

Our simulation here closely resembles that of Example 1, when excluding the confounding. The DGP without $U$ is as follows:

$$
\begin{aligned}
\varepsilon_Y &\sim \mathcal{N}(0, 1) \\
Z &\sim \mathcal{N}(-2, 1) \\
X &\sim \mathcal{N}(-4, 5) \\
S &:= \mathbf{1}_{\{X+Z<-6\}} \\
Y &:= 0.2X^2 + 5Z + \varepsilon_Y
\end{aligned}
$$

$$
E[Y \mid X = x] = 0.2x^2 - 2 + 2x
$$
$$
E[Y \mid do(X = x)] = 0.2x^2 - 10
$$

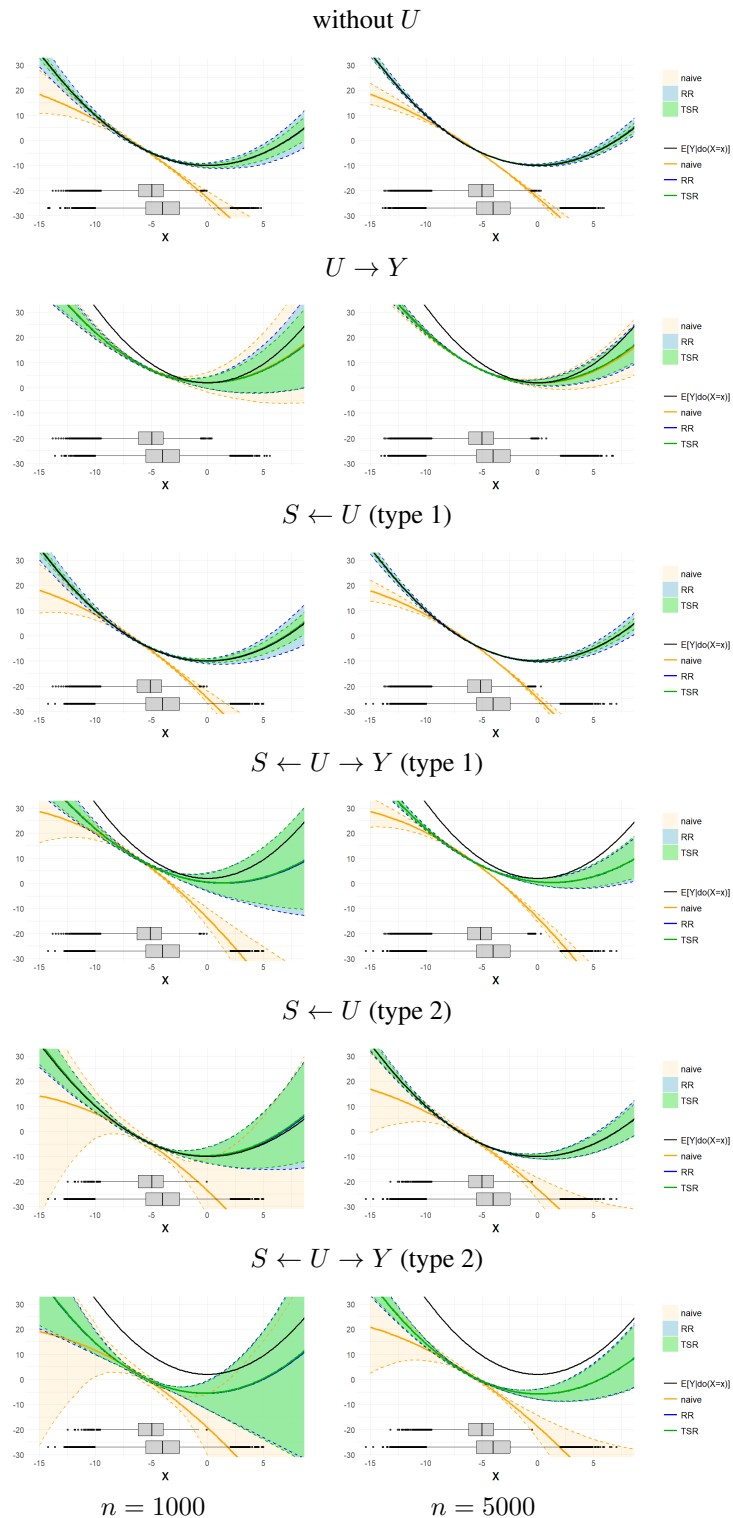

Figure 18: Comparison of the central 95%-areas of RR, TSR and the naive estimator of the simulation runs in absence of $U$ as well as introducing $U \rightarrow Y$, $S \leftarrow U$ or $S \leftarrow U \rightarrow Y$ with sample size $n \in \{1000, 5000\}$. The upper boxplot represents the distribution of $X$ in $\mathcal{S}$ and the lower in $\mathcal{D}$ ($\mathcal{S} \cap \mathcal{D} = \emptyset$). The curves for RR, TSR and the naive estimator display the mean estimation over the simulation runs.

For the three cases described above, we include $U \sim \mathcal{N}(3, 1)$. Below we state how the DGP is extended for the three cases, respectively. For the edge $S \leftarrow U$, we distinguish between two cases.

(a) $U \rightarrow Y$
$Y := 0.2X^2 + 5Z + 4U + \varepsilon_Y$

(b) $S \leftarrow U$
$S := \mathbf{1}(X + Z + 0.1U < -5)$ (type 1)
$S := \mathbf{1}(X + Z < -5)\mathbf{1}(U < 1.5)$ (type 2)

(c) $S \leftarrow U \rightarrow Y$
$S := \mathbf{1}(X + Z + 0.1U < -5)$ (type 1)
$S := \mathbf{1}(X + Z < -5)\mathbf{1}(U < 1.5)$ (type 2)
$Y := 0.2X^2 + 5Z + 4U + \varepsilon_Y$

We followed the procedure described in Section 4 and included covariates up to degree 2 in the regression model. In this analysis, we restricted ourselves to OLS regression. The results, given in Figure 18 are consistent with what we expected based on our previous considerations. In case (a) a bias arises from the missing regressor (see Figure 18). For case (b), our assumptions are not violated. Hence, TSR yields reliable results. For type 2, we just need to increase the sample size $n$ to compensate the decrease of sample size of the selected sample induced by adding $S \leftarrow U$. For case (c), the estimation gets biased due to the inclusion of an unobserved regressor. In addition, the specification of TSR is wrong as the derivation exploits PMAR. For cases (a) and (b), where their respective assumptions are satisfied, TSR again shows a smaller variance than RR.

