# OpenReview forum: "Regression-Based Estimation of Causal Effects in the Presence of Selection Bias and Confounding"
_NeurIPS.cc/2025/Workshop/Reliable_ML — NeurIPS 2025 - Reliable ML Workshop_

### Official Review · Reviewer_nxux · 2025-09-15
**Regression-Based Estimation of Causal Effects in the Presence of Selection Bias and Confounding**

**Rating:** 9
**Confidence:** 3

**Review:**

# Summary
Authors consider estimation of causal effect when the dataset is conditioned on the positive result of a selection (dependent on variables X) and suffers from confounding. Using access to external data, collected outside the selection mechanism, authors show that the causal effect is identifiable and s-recoverable. They then derive a Two-Step Regression Estimator (TSR) for this effect. Theoretical results indicate TSR should yield a lower mean-squared error (MSE) than Repeated Regression (RR), which is supported by the empirical simulations.

# Strengths
1. Authors present both identification and estimation results for causal effect estimation under an expanded setting from existing work in Boeken et al.
2. Introduction and set-up very clearly describes the underlying model and its relation to existing work. Appendix even further compares the assumptions to prior settings.
3. Experiments complement theoretical results and further present nonlinear results.

# Weaknesses/Limitations
1. The authors may consider adding citations (and perhaps a discussion to the appendix) about the relevance and technical relation of classic econometrics methodologies to estimate causal effects under unmeasured confounding particularly Two-Stage Least Squares (2SLS).

# Suggestions to Authors
The paper is well-written and structurally easy to follow. In addition to the suggested discussion about 2SLS above, some citations to estimation under model selection, e.g. “What Makes a Good Fisherman? Linear Regression under Self-Selection Bias” (Cherapanamjeri et al) and referenced works, thought these are understandably less relevant as the setting involves selection such that $S \in \{1, …k\}$ rather than $S \in \{0,1\}$

---

### Official Review · Reviewer_yHqP · 2025-09-19
**Strong generalization under strong assumptions.**

**Rating:** 6
**Confidence:** 2

**Review:**

Summary:
The authors address the issue of estimating the expected causal effect, E[Y|do(X)], of a target variable, Y, when the treatment, X, is determined by an intervention. The authors focus on continuous random variables. They focus on identification under selection bias in the presence of confounding.  They derive theoretical conditions that ensure the identifiability and recoverability of causal effects when external data and proxy variables are available. They introduce a two-step regression estimator, analyze its performance, provide theoretical guarantees, and experimentally validate its correctness.
Strengths:
1)The authors emphasize the importance of the problem. The problem lies within the scope of the workshop.
2)The authors derived sufficient conditions under which causal effects are identifiable and s-recoverable.
3)They introduced a generalized estimator that matches or outperforms previous estimators, which were built to address only a subset of the settings.
Weaknesses:
1)The assumptions used by the authors are strong.
2) (Minor) They only used synthetic data to evaluate their estimator's performance and compare it to repeated regression.
3) (Minor) Although the phrasing is excellent and the writing is generally clear, a significant portion of the paper, including most of the proofs, is contained in the appendix. Thus, the main text is not self-sufficient.
Suggestions:
Following the directions in the Future Work paragraph at the end of the "Conclusions" section seems very promising.
In my subjective opinion, you should structure the paper's writing around the comparison with RR less, as it makes your paper feel incremental. Obviously, mention RR, but possibly mention it less.

---

### Official Review · Reviewer_TCg7 · 2025-09-21

**Rating:** 8
**Confidence:** 4

**Review:**

### Summary

The paper studies estimation of the **average causal effect** of an intervention $X$ on an outcome $Y$, i.e.,
$
\mathbb{E}[Y \mid \mathrm{do}(X)],
$
under **selection bias** and **confounding**.
Selection bias here means there exists a variable $S \in \{0,1\}$ and we observe $Y$ only when $S=1$.
Confounding exists when there is a set $Z$ that influences both $X$ and $Y$, affecting their causal relationship.

Prior work studied the problem under only selection bias and proposed an estimator called **Repeated Regression (RR)**. This work extends those results by proposing an estimator for the case where confounding might also exist—the **Two-Step Regression (TSR)** estimator.  They also show that TSR achieves reduced variance compared to RR in the setting where only selection bias is assumed.

As is common in this literature, the estimator succeeds under assumptions about the causal graph structure and the available information in the data (what they call *proxy variables* and *external data*).  Specifically, they assume access to a set $Z$ that, along with $X$, blocks all paths between $S$ and $Y$, so there is no selection bias after conditioning on $(Z,X)$. Moreover, they assume access to a set of variables $Z^+$ which blocks all the back-door paths from $X$ to $Y$, so there is no confounding after conditioning on $Z^+$.

The authors support their findings with experiments in two settings: (i) $Y$ is linear in $X$ and $Z$; and (ii) $Y$ is quadratic in $X$ and linear in $Z$.

---

### Strengths

- The paper is well written, states the problem clearly, and compares it with related work. It addresses an important problem for both theory and practice.
- It substantially improves prior work by extending findings to the case where confounding exists between the cause $X$ and effect $Y$.
- The proposed estimator is simple and easy to implement.
- The authors provide extensive experiments to back their results.

---

### Weaknesses

- The correctness of the estimator relies on *knowing* the causal relationships among variables and having access to all the variables that affect selection bias and confounding.
- Related to the above, the last experiment claims that TSR can retrieve the causal effect even when hidden covariates exist. This can be misleading: the result works *because* the observed $Z$ already blocks every back-door path from $X$ to $Y$. Without such a $Z$, hidden confounders would still bias the effect.
- The variance gap between RR and TSR has an easy fix in the presented no-confounding setting: if $Z \perp X$, then $\mathbb{E}[Z \mid X] = \mathbb{E}[Z]$. The variance of RR is caused because estimating $\mathbb{E}[Z \mid X]$ from data introduces extra error. A careful RR implementation could simply estimate $\mathbb{E}[Z]$ and avoid the excess variance.
- The paper does not comment on efficiency/runtime or the computational assumptions behind the method.

---

### Suggestions

The authors mention they already work on relaxing their assumptions on the available data and get a more robust estimator for the problem. I beleive this is an important step that would further strengthen their results.

The separation between the variance of RR and TSR estimators through the specific example is not so imposing. Is there any other example where $X, Z$ are not independent and the separation still holds?

---

### Overall

The paper proposes a new estimator for average causal effects in the presence of selection bias and confounding. It addresses a fundamental problem for both theory and practice and constitutes an important addition to prior work.